# REINFORCEMENT SYMBOLIC REGRESSION MACHINE

**Yilong Xu[1], Yang Liu[2], Hao Sun[1,*]**
[1]Gaoling School of Artificial Intelligence, Renmin University of China, Beijing, China;
[2]School of Engineering Science, University of Chinese Academy of Sciences, Beijing, China;
Emails: `xuyilong88@ruc.edu.cn; liuyang22@ucas.ac.cn; haosun@ruc.edu.cn`

## ABSTRACT

In nature, the behavior of many complex systems can be described by parsimonious math equations. Symbolic Regression (SR) is defined as the task of automatically distilling equations from limited data. Keen efforts have been placed on tackling this issue and demonstrated success in SR. However, there still exist bottlenecks that current methods struggle to break, when the expressions we need to explore tend toward infinity and especially when the underlying math formula is intricate. To this end, we propose a novel Reinforcement Symbolic Regression Machine (RSRM) that masters the capability of uncovering complex math equations from only scarce data. The RSRM model is composed of three key modules: (1) a Monte Carlo tree search (MCTS) agent, designed for exploration, that explores optimal math expression trees consisting of pre-defined math operators and variables, (2) a Double Q-learning block, designed for exploitation, that helps reduce the feasible search space of MCTS via properly understanding the distribution of reward, and (3) a modulated sub-tree discovery block that heuristically learns and defines new math operators to improve representation ability of math expression trees. Binding of these modules yields the SOTA performance of RSRM in SR as demonstrated by multiple benchmark datasets. The RSRM shows clear superiority over several representative baseline models.

## 1 INTRODUCTION

The pursuit of mathematical expressions through data represents a crucial undertaking in contemporary scientific research. The availability of quantitative mathematical expressions to depict natural relationships enhances human comprehension and yields more precise insights. Parsing solutions offer superior interpretability and generalization compared to numerical solutions generated by neural networks. Additionally, simple expressions exhibit computational efficiency advantages over the latter. As a result, these techniques have found applications across diverse fields, e.g., discovering fundamental physical laws (Udrescu & Tegmark, 2020; Liu & Tegmark, 2021) or governing equations (Schmidt & Lipson, 2009; Chen et al., 2021; Sun et al., 2023), modeling material constitutive relations (Wang et al., 2019), and TCP congestion control Sharan et al. (2022), among many others.

The process of fitting expressions in early years involves polynomial interpolation to derive an equation, followed by the appearance of the SINDy method (Kaiser et al., 2018), which utilizes sparse regression to identify appropriate mathematical expressions based on a predefined library of candidate terms. These methods e.g., (Sun et al., 2021; Chen et al., 2021; Champion, 2019), effectively reduce the search space from an infinitely large set of possibilities to a limited fixed set of expressions, thereby narrowing down the search process. However, the applicability of this approach is limited, since the compositional structure of many equations cannot be predefined in advance. Therefore, there is a need for more comprehensive methods to search for expressions.

The Equation Learner (Martius & Lampert, 2016; Sahoo et al., 2018) model was then introduced as a novel method in symbolic learning, which incorporates symbolic operators as activation functions. This modification enabled the neural network to generate more precise and interpretable functional relationships, allowing for the discovery of intricate math expressions. However, given the compact structure of EQL, optimizing the sparse network to distill parsimonious equations is a key challenge.

Another approach involves generating optimal expression trees (Hopcroft et al., 2006), where internal nodes correspond to operators and each leaf node represents a constant or variable. By recursively

---

*Corresponding author

computing the expressions of the sub-trees, these expression trees can be transformed into math expressions. Initially, genetic programming (GP) (Schmidt & Lipson, 2009; Augusto & Barbosa, 2000; Gustafson et al., 2005) was employed to address these problems. Although GP showed promise, its sensitivity to parameter settings leads to instability. Deep learning methods emerged then to tackle the problem. SymbolicGPT (Valipour et al., 2021) utilizes a generative model like GPT to create expression trees, while AIFeynman (Udrescu & Tegmark, 2020) uses neural networks to analyze the relationships and dependencies between variables and search for relevant expressions. Despite its ad-hoc characteristic, the AIFeynman (Udrescu et al., 2020) method was further improved, offering faster and more precise expression search capabilities. Additionally, reinforcement learning (RL) (Sun et al., 2023) has been employed, which utilizes the Monte Carlo tree search method to explore and discover expressions, along with a module-transplant module that generates new expressions based on existing ones. Deep (RL) methods, e.g. DSR (Petersen et al., 2019), utilize recurrent neural networks to learn expression features and generate probabilities. A policy gradient search algorithm samples the probabilities to generate a batch of expressions, which are subsequently evaluated for performance. Combining DSR and GP leads to a new model called NGGP (Mundhenk et al., 2021a), which achieves better performance. Then uDSR (Landajuela et al., 2022), a comprehensive framework that combines DSR, AIFeynman, LSPT (Large-scale pre-training), GP, and LM (Linear models) emerged to enhance the efficiency and accuracy of symbolic regression. Pre-trained generative models (Holt et al., 2022) and end-to-end transformer modules (Kamienny et al., 2022; Li et al., 2022) also achieved satisfactory expression search results.

Nevertheless, the existing methods still struggle with generating lengthy and complex equations, and are faced with issues related to overfitting, e.g., poor generalizability. To overcome these challenges, we propose a model named Reinforcement Symbolic Regression Machine (RSRM) that masters the capability of uncovering complex math equations from only scarce data, composed of an RL-search agent, a GP-based expression tuning element, and a modulated sub-tree discovery (MSDB) block. The RL-search agent is designed based on the synergy between Monte Carlo tree search (MCTS) (Coulom, 2006) and double Q-learning (Hasselt, 2010) for enhanced exploration and exploitation. The GP learner is employed to fine-tune the generated expression trees (e.g., see the demonstration in (Mundhenk et al., 2021a)), while the (MSDB) block heuristically learns and defines new math operators to improve the representation ability of math expression trees.

We would like to emphasize that MSDB addresses a crucial observation that models often struggle to generate complete expressions but excel in capturing certain components. For instance, NGGP (Mundhenk et al., 2021a) may discover an expression like $x^4 - x^3 + \cos(y) + x - 1$, while the ground truth is $x^4 - x^3 - 0.5y^2 + x$. Notably, it successfully recovers the simplified expression $x - 0.5y^2$ with the same distribution. To this end, MSDB offers a new alternative to simplify expressions by subtracting specific components in the context of a sub-tree, as exemplified by the subtraction of $x^4 - x^3$ in the aforementioned case. Such an MSDB module takes the divide-and-conquer concept and could significantly improve the overall search performance of the RSRM model.

The aforementioned aspects form the main **contributions** of this paper: Our proposed RSRM model offers a novel solution to the search for mathematical expressions. By incorporating double Q-learning into MCTS, we effectively balance exploration and exploitation of SR tasks. The proposed (MSDB) block can handle equations with symmetry (reducing the complexity), and assist in dealing with long equations by identifying common patterns and defining new math operators on the fly. As a result, the RSRM model demonstrates clear superiority over several baseline models, which surpasses that of the baseline models in terms of accuracy and generalization ability.

## 2 BACKGROUND

**Genetic Programming:** Genetic programming (Stephens, 2016; Koza, 1994; Schmidt & Lipson, 2009) is employed to iteratively improve expression trees in order to approximate the optimal expression tree. The mutation step in GP enables random mutations in the expression tree, while genetic recombination allows for the exchange of sub-trees between expression trees, leading to the creation of new expression trees based on the knowledge acquired from previous generations. This "genetic evolution" process progressively yields highly favorable outcomes after a few generations.

**Double Q-Learning:** Double Q-learning (Hasselt, 2010) is a reinforcement learning algorithm designed to overcome the overestimation bias issue in traditional Q-learning. The key idea behind Double Q-learning is to use two sets of Q-values to independently estimate the value of each action in

a given state. By using two separate Q-functions, Double Q-learning can mitigate the overestimation bias of traditional Q-learning and provide more accurate value estimations, leading to better policy learning and performance in various reinforcement learning tasks.

**Monte Carlo Tree Search:** MCTS (Coulom, 2006) is a decision-making search algorithm that constructs a search tree representing possible game states and associated values. It employs stochastic simulations to explore the tree and determine the value of each node. This algorithm gained prominence via its adoption by the AlphaZero team (Silver et al., 2017). MCTS consists of four steps in each iteration: (1) selection, (2) expansion, (3) simulation, and (4) backpropagation. During selection, the best child node is chosen based on certain criteria. If an expandable node lacks children, it is extended by adding available children. The simulation step involves simulating the current state before selecting the next node, often using the Upper Confidence Bound for Trees (UCT) algorithm to calculate the selection probabilities, defined as $UCT(v') = \bar{Q}(v') + c\sqrt{\ln(N(v))/N(v')}$. Here, $\bar{Q}(v')$ means the average reward of the child node, $N(v)$ and $N(v')$ represents the number of visits to the current node and its child node, respectively, and $c$ typically represents the exploration-exploitation tradeoff parameter. The first part of the equation makes the nodes with high reward visit more often, and the second part ensures that the nodes with fewer visits have a higher probability of being selected. Finally, in the backpropagation step, the reward function evaluates child nodes, and their values are used to update the values of parent nodes in the tree. The theoretical analysis (e.g., convergence, guarantees)) of the UCT-based MCTS algorithm can be found in Shah et al. (2022).

## 3 METHOD

The RSRM model consists of a three-step symbolic learning process: RL-based expression search, GP tuning, and MSDB. With these steps, our model effectively learns and represents the relationship present in the data, facilitating accurate and interpretable modeling. The schematic representation of RSRM is depicted in Figure 1. The full settings of our model are in Appendix A.

The RL search consists of a double Q-learning empowered MCTS agent. Here, MCTS is employed for exploration (global search) that aids in generating unexplored expressions, while double Q-learning enables exploitation that captures the local distribution of equations. Additionally, we adopt a method that involves visiting each child node a specific number of times before activating double Q-learning. This approach aims to avoid excessive reliance on historical information, mitigating the

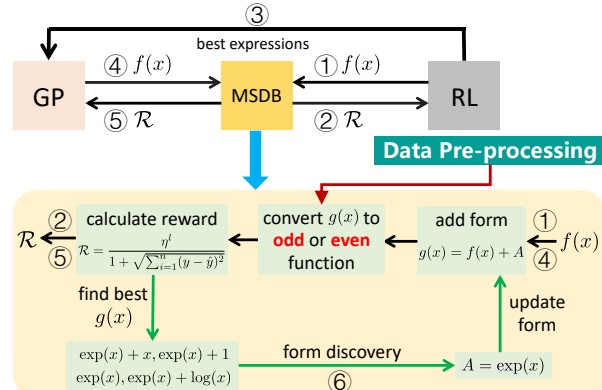

Figure 1: RSRM comprises several steps. First, the data is input into the data pre-processing module to determine the parity. Each epoch, the process initiates with the utilization of the MCTS module to generate expressions (Step ①). Subsequently, these expressions undergo the addition of forms and reward calculation (Step ②). The expressions are then refined through GP (Step ③), leading to more accurate expressions (Step ④). Following this, the equation forms are integrated and rewards are calculated (Step ⑤). This iterative cycle, involving MCTS and GP, continues for several rounds, ultimately resulting in the renewal of forms based on the most proficient expressions (Step ⑥).

risk of overfitting and promoting a more robust learning process. To address the challenge of lengthy and hard equations, we introduce an interpolation method (e.g., data pre-processing) to identify whether the equation exhibits symmetry prior to each search, followed by a modulated sub-tree discovery block (MSDB). If symmetry is present, we pre-process the equation accordingly to simplify the subsequent search process. This approach effectively reduces the difficulty associated with specific equations. The MSDB examines whether the few expressions that perform well adhere to a specific form. This divide-and-conquer algorithm enables a step-by-step search for equations, facilitating the generation of long expressions.

### 3.1 EXPRESSION TREE

The objective of SR can be transformed into the generation of an optimal expression tree (Hopcroft et al., 2006), which represents a mathematical expression. The expression tree consists of *internal*

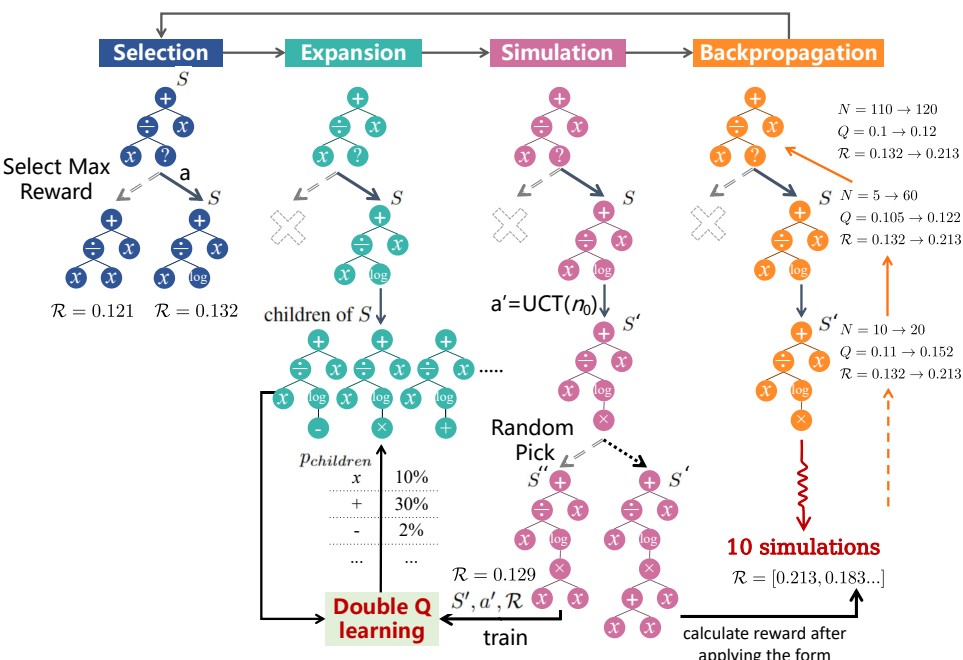

Figure 2: Schematic of the proposed RL search. MCTS selects functions based on the maximum reward, expands them using the results of double Q-learning, simulates node selection through the UCT function, randomly fills the current tree, and provides rewards to double Q-learning to train. Once the generation is complete, the rewards are back-propagated to the parent node.

*nodes* that correspond to operators (e.g., $+, -, \times, \div, \log, \exp, \sin, \cos$) and *leaf nodes* that correspond to constants (e.g., $1, 2$) or variables (e.g., $x$). By recursively computing the expressions of the sub-trees, the expression tree can be transformed into mathematical expressions. The process of generating an expression tree follows a recursive method where operators are added until no more can be added. This approach simplifies the task of creating expressions as it focuses on constructing the expression tree, which can be easily generated using recursive techniques.

In contrast to previous methods, we employ a hierarchical traversal strategy for generating expression trees. This is motivated by the Monte Carlo tree search algorithm, where conducting more searches on vertices that are filled earlier is deemed more beneficial. In the context of constructing expression trees, this implies that higher-level nodes in the tree carry greater significance. Consequently, We use a hierarchical construction method to build the expression tree layer by layer, similar to the hierarchical traversal of trees.

### 3.2 REINFORCEMENT LEARNING GUIDED SEARCH

The search step relies on the double Q-learning and MCTS algorithms, which are shown in Figure 2. The specific algorithm is shown in Algorithm 1.

**Reward function**: The reward function used in our approach is based on the root mean square error (RMSE) and is designed to evaluate the fit of the generated equations to the measured data. It promotes concise and accurate expressions by assigning higher rewards to shorter and more precise functions. Inspired by the SPL approach (Sun et al., 2023), the reward function is computed by:

$$\mathcal{R} = \frac{\eta^l}{1 + \sqrt{\sum_{i=1}^{n}(y_i - \hat{y}_i)^2}}, \tag{1}$$

where $\eta$ is a discount factor promoting concise trees, and $l$ is the number of nodes in the expression tree. $y_i$ and $\hat{y}_i$ the true and the predicted values generated by the MSDB with the output of Reinforcement Learning Search of the $i$th data point, respectively. Using this reward function, our approach encourages the discovery of equations that minimize the RMSE and favors shorter and more concise expressions, leading to higher reward values for functions that provide better fits to the data.

---

**Algorithm 1** Expression generation by RSRM

---

**Input:** dataset $\mathcal{S}_{data}$, expression form $\mathcal{F}$
**Parameters:** discount rate $\eta$, UCT const $c$, minimum selected times $n_0$
**Outputs:** best expression
  Initiate $S$ as top of MTCS
  **Selection**:
  $a \leftarrow$ children of $S$ with maxium $\mathcal{R}$                                  ▷ Greedy selection
  $S$ take action $a$
  **Simulation**:
  $S' \leftarrow S$
  **repeat**
      **if** children of $S$ is empty **then** Expand $S'$
      **end if**
      **if** $\exists x \in$ children of $S' \rightarrow (N(x) < n_0)$ **then** $a' \leftarrow x$       ▷ Select child with visit times $< n_0$
      **else**  $a' \leftarrow$ randomly choose child of $S'$ by $UCT$         ▷ Select through UCT function
      **end if**
      $S'$ take action $a'$, $S'' \leftarrow S'$, Fill up randomly $S''$
      double Q-learning $\leftarrow S', a', \mathcal{R}$ of $S''$          ▷ train double Q-learning by simulated reward
  **until** $S'$ is full
  **Expansion**
  children of $S \rightarrow$ double Q-learning $\rightarrow p_{children}$         ▷ estimate initial possibility of each child
  **Back-propagate**
  **Back-propagate** $\mathcal{R}$ of $S'$ based on $\mathcal{F}$

---

**Greedy selection**: Our method employs greedy selection, similar to Sun et al. (2023). Instead of selecting the token with the highest UCT score, we choose the token that currently yields the best reward (Eq. 1). This ensures the selection of tokens leading to expressions resembling the current best one, potentially resulting in improved expressions, yet, increasing the possibility of overfitting. Note that UCT is employed during the MCTS simulation while the greedy selection of the maximum reward is applied to choose the optimal expression tree.

**Simulated reward**: At each token generation, the entire expression tree is randomly completed based on the current tree. The reward is then computed using the reward function and fed back to double Q-learning for training. This approach avoids excessive rounds of learning at the top node and filters out irrelevant nodes initially.

**Parameter optimization**: After an expression tree is built, we need to fill the parameter (i.e., equation coefficients) placeholders in it. We treat each placeholder as an unknown variable, which is optimized to maximize the reward. The BFGS (Roger Fletcher & Sons, 2013) algorithm, available in the scipy (Virtanen et al., 2020) module in Python, is used for optimization. In contrast to the approach in DSR (Petersen et al., 2019), we find that Gaussian random numbers with a unit mean and variance provide more effective initial values for optimization (see further information in Appendix Section C.6 where we test the performance of the model with different initial values).

## 3.3 MODULATED SUB-TREE DISCOVERY

We incorporate three specific sub-tree expression forms to enhance the exploration and analysis of equations, where $\mathcal{A}$ represents a fixed form and $f(x)$ a learnable part, explained as follows:

- $\mathcal{A} + f(x)$: This search form focuses on identifying expressions of the form like $e^x - x$ and $e^x + x$. By recognizing this pattern, we can effectively explore and analyze equations that follow the structure of $e^x + f(x)$.

- $\mathcal{A} \times f(x)$: In this search form, we obtain good expressions such as $1.57e^x$ and $1.56e^x + x$, aiming to detect equations of the form $e^x \times f(x)$.

- $\mathcal{A}^{f(x)}$: The search form $\mathcal{A}^{f(x)}$ is designed to recognize equations like $(e^x)^{2.5}$ and $(e^x)^e$, indicating the presence of expressions in the form $(e^x)^{f(x)}$.

Our approach involves the establishment of these forms based on the initial token of the expression tree, because the root of an expression tree serves as a focal point, indicating the primary operation or function in the expression. Thus, we separate the sub-tree forms based on it. Specifically, if the first

token corresponds to addition ($+$) or subtraction ($-$), the method proceeds to learn the generation of the left and right sides of the respective operators. Similarly, for tokens such as multiplication ($\times$), division ($\div$), or exponentiation ($\wedge$), a similar procedure is followed. In the case of unary expressions, such as trigonometric functions ($\sin$ and $\cos$), the MCTS and GP models effortlessly derive the complete expression. Therefore, while our method involves a degree of empirical design in identifying the sub-tree expression forms, it possesses a universal nature.

The complete form-discovery algorithm, which outlines the procedure for selecting and generating the search form among the three options, is provided in Algorithm 2 and Appendix Figure S1.

---

**Algorithm 2** Search for the form of the expression through the generated expressions

---

**Input:** best expression set $\mathcal{S}_{best}$
**Parameters:** selection ratio $k_s$, expression percentage ratio $k_p$, maximum select number $N$
**Output:** the form of the expression $\mathcal{F}$

   $l \leftarrow$ length of $(\mathcal{S}_{best})$
   Sort $\mathcal{S}_{best}$ by $\mathcal{R}$ decent
   **for** $i$ in $1, 2...l$ **do**
      **if** $i \leq N$ and $\mathcal{R}(\mathcal{S}_{best}[i]) \geq k \times \mathcal{R}_{max}$ **then**          $\triangleright$ If number of $\mathcal{G}$ exceeds or $\mathcal{R}$ is low, break out
         $\mathcal{D} = \mathcal{D} + \text{Split}(\mathcal{S}_{best}[i])$     $\triangleright$ Occurrences of function in Split-by-addition($\mathcal{S}_{best}[i]$) $+1$
      **end if**
   **end for**
   $\mathcal{G}_0 \leftarrow \mathcal{D}$ with maximum number of occurrences.
   **if** $\exists\mathcal{C} \notin Z \rightarrow \mathcal{G}_0 = \mathcal{A}^{\mathcal{C}}$ **then** $\mathcal{F} = \mathcal{A}^{f(x)}$          $\triangleright$ The form is $\mathcal{A}^{f(x)}$, $Z$ means integer set.
   **else if** $\exists\mathcal{C} \notin Z \rightarrow \mathcal{G}_0 = \mathcal{A} \times \mathcal{C}$ **then** $\mathcal{F} = \mathcal{A} \times f(x)$          $\triangleright$ The form is $\mathcal{A} \times f(x)$
   **else**
      $\mathcal{F} = f(x)$          $\triangleright$ The form is $\mathcal{A} \pm F(x)$
      **for** $\mathcal{G}$ in $\mathcal{D}$ **do**
         **if** Occurrences of $\mathcal{G} \geq l \times k_p$ **then** $\mathcal{F} = \mathcal{F} + \mathcal{G}$          $\triangleright$ Add $\mathcal{G}$ to $\mathcal{A}$
         **end if**
      **end for**
   **end if**

---

**Splitting by Addition:** In this step, we convert the formula, which is represented as a token set, into a string using a library like sympy (Meurer et al., 2017). Then we expand the expression into a sum of simpler expressions. Next, we split the expanded expression into multiple simple expressions using sum or difference notation. In this way, we convert $[+, \times, -, x, y, z, \log, x]$ to $x \times y + z - \log(x)$, and then transforms it to $xy, z, -\log(x)$.

Once the expression is refined into its desired form, the subsequent search becomes more manageable. For instance, in the case of aiming to derive $\exp(x^2) + x^4 + x^3 + 0.5\log(x)$, we can break down the search. Initially, we generate $\exp(x^2) + ...$ to identify the form $\exp(x^2) + f(x)$, and then extend this to $\exp(x^2) + x^4 + x^3 + f(x)$, simplifying the process of obtaining $\exp(x^2) + x^4 + x^3 + 0.5\log(x)$.

Inspired by the approach proposed by Udrescu & Tegmark (2020), we introduce a data pre-processing module to determine the potential parity of the underlying equation. The cubic splines (Catmull & Rom, 1974) are applied for equation fitting, generating a function. Subsequently, this function is used to compute the relationship between $y(-x)$ and $y(x)$, enabling the determination of whether $y(x)$ is an odd, even, or neither function, where $y(x)$ is the relation of $x$ and $y$ in given data.

When the error (RMSE) between $y(-x)$ and $y(x)$ remains below constant $E_{sym}$, the function is considered even with respect to $x$. Negative values of the independent variables are transformed to their absolute values, while retaining the dependent variable values. Further exploration is conducted using the form of $\hat{y} = (g(x) + g(-x))/2$ to make it discover specific forms.

Similarly, if the error between $y(-x)$ and $y(x)$ is within the limit constant $E_{sym}$, the function is classified as odd relative to $x$. Negative values of the independent variables are converted to absolute values, and the dependent variable values are inverted. The search continues employing the form of $\hat{y} = (g(x) - g(-x))/2$ to make it odd in discovering specific forms.

Once the expression is refined to its parity form, the difficulty of searching for the expression is reduced. If we want to get $\cosh(x)$, we only need to generate $\exp(x)$ after parity determination. Such a partitioning strategy has been used in the past, e.g., Petersen et al. (2019) using sub-trees as new tokens, Sun et al. (2023) using transplanted sub-trees, Udrescu & Tegmark (2020) using problem

Table 1: Recover rate (%) of several difficult equations in symbolic regression: trigonometric functions and sum of multiple power functions with parameter $1/2$ in Nguyen; power functions and trigonometric functions in Nguyen$^c$, trigonometric functions and hyperbolic function and functions with weird power in Livermore; rational functions in R and R$^0$ (where $x = 0, y = 0$ add to dataset).

| BenchMark | Equation | Ours | SPL | uDSR | NGGP | DSR | GP |
|---|---|---|---|---|---|---|---|
| Nguyen-5 | $\sin(x_1^2)\cos(x_1) - 1$ | **100** | 95 | 55 | 80 | 72 | 12 |
| Nguyen-12 | $x_1^4 - x_1^3 - 0.5x_2^2 + x_2$ | **100** | 28 | 30 | 21 | 0 | 0 |
| Nguyen-2$^c$ | $0.48x_1^4 + 3.39x_1^3 + 2.12x_1^2 + 1.78x_1$ | **100** | 94 | **100** | 98 | 90 | 0 |
| Nguyen-9$^c$ | $\sin(1.5x_1) + \sin(0.5x_2^2)$ | **100** | 96 | 0 | 90 | 65 | 0 |
| Livermore-3 | $\sin(x_1^3)\cos(x_1^2) - 1$ | **55** | 15 | 0 | 2 | 0 | 0 |
| Livermore-7 | $\sinh(x_1)$ | **100** | 18 | 0 | 24 | 3 | 0 |
| Livermore-16 | $x_1^{2/5}$ | **100** | 40 | 60 | 26 | 10 | 5 |
| Livermore-18 | $\sin(x_1^2)\cos(x_1) - 5$ | **100** | 80 | 59 | 33 | 0 | 0 |
| AIFeynman-9 | $x_1 + x_2 + 2\sqrt{x_1 x_2}\cos(x_3)$ | **67** | 0 | 8 | 7 | 0 | 0 |
| AIFeynman-10 | $\frac{1}{2}x_1(x_2^2 + x_3^2 + x_4^2)$ | **15** | 0 | 0 | 0 | 0 | 0 |
| R-1$^0$ | $(x_1 + 1)^3/(x_1^2 - x_1 + 1)$ | **49** | 0 | 17 | 2 | 0 | 0 |
| R-2$^0$ | $(x_1^5 - 3x_1^3 + 1)/(x_1^2 + 1)$ | **89** | 0 | 0 | 0 | 0 | 0 |
| R-3$^0$ | $(x_1^5 + x_1^6)/(x_1^4 + x_1^3 + x_1^2 + x_1 + 1)$ | **91** | 0 | 0 | 4 | 0 | 0 |

decomposition. Our main innovation is the use of partial sub-trees separated according to the plus sign as part of the new expression.

Also, we conduct a parity determination performance test to compare the efficiency and effectiveness of form discovery by AIFeynman and our method. The experiment setting and results are given in Appendix G. It shows that our method (based on cubic splines) outperforms AIFeynman (based on MLP) in terms of higher accuracy and smaller data requirements.

## 4 RESULTS

We test the performance of our method on multiple different datasets and compare it with the following baseline models in symbolic learning: SPL (Sun et al., 2023), DSR (Petersen et al., 2019), NGGP (Mundhenk et al., 2021a), uDSR (Landajuela et al., 2022), DGSR (Holt et al., 2022), gplearn (Stephens, 2016), and AFP-FE (Schmidt & Lipson, 2010). The description of each baseline along with parameter setting is found in Appendix B.

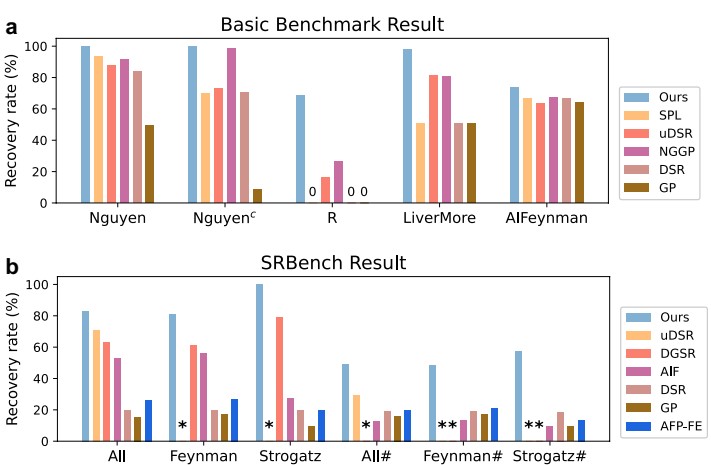

Figure 3: Recover rates of benchmark datasets. **a**, Basic benchmarks (detailed results shown in Appendix C). **b**, SRbench dataset (detailed results found in Appendix D and Appendix Figure S3), where the symbol # denotes the presence of noise with a mean of $10^{-3}$ added to the target values and ∗ represents missing data in the literature.

### 4.1 BASIC BENCHMARKS

To evaluate the efficiency of our model, we first utilize four basic benchmark datasets (see Appendix C for details): Nguyen (Uy et al., 2011), Nguyen$^c$ (McDermott et al., 2012), R (Mundhenk et al., 2021b), Livermore (Mundhenk et al., 2021b), and AIFeynman (Udrescu & Tegmark, 2020). Note that parameter optimization (e.g., calibration of the equation coefficients) is prohibited in this experiment except the Nyugen$^c$ dataset.

We employ the recovery rate as the evaluation metric, which measures the number of times the correct expression is recovered across multiple independent repetitions of a test. Note that this metric ensures that the model's output exactly matches the target expression.

We summarize the comparison of recovery rates for several difficult expressions on the five benchmark datasets listed in Table 1. The results demonstrate that our model performs well on these complex expressions. Furthermore, we compared the mean recovery rates of all equations on each benchmark (see Figure 3a and Appendix C). Our method outperforms other approaches, achieving the highest recovery rates for all benchmark expressions.

We also conducted an experiment on the trade-off between accuracy and the number of evaluations in the Nyugen Benchmark. Details of this experiment are given in Appendix Section C.7.

## 4.2 SRBENCH DATASET

We further tested our model's ability to learn more complex equations with noisy training data using the SRbench dataset La Cava et al. (2021), where parameter optimization is allowed. This dataset comprises 252 datasets sourced from Romano et al. (2021). We specifically concentrated on 131 of these datasets equipped with ground truth equations, which were drawn from two primary sources, namely, AIFeynman (Udrescu & Tegmark, 2020) and Strogatz (Strogatz, 2014) datasets.

Our evaluation encompassed a spectrum of baseline models from SRBench, as well as more contemporary approaches like uDSR (Landajuela et al., 2022) and DGSR (Holt et al., 2022). The recovery rate results of each method for different testing datasets (e.g., all datasets, only Feynman, only Strogatz) and noise effect can be found in Figure 3b. Here, the results for the AIfeynman dataset in Figure 3b are inconsistent with Figure 3a since parameter optimization is prohibited in a but engraved in b. Notably, our MSDB module demonstrated proficiency in handling intricate equations, such as the challenging example $-(32x_1^4x_3^2x_4^2(x_3 + x_4))/(5x_2^5x_5^5)$, which can be effectively discovered through the form $(Cx_1^4)/(x_2^5x_5^5) \times f(x)$. Additionally, numerous equations, like $-10/3x_1^3 - 10/3x_1 + 10x_2$, were successfully identified by aggregating multiple smaller equations, a feat achievable through the form $Cx_1^3 + f(x)$. Consequently, our model exhibited proficiency in uncovering a wide range of equations. The comparative analysis in Figure 3b clearly illustrates the superior performance of our approach in comparison with other baseline models.

We further tested our model by discovering a surrogate formula to approximate the cumulative density function (CDF) of a normal distribution, e.g., $F(x, \mu, \sigma) = \int_{-\infty}^{x} \frac{1}{\sqrt{2\pi}\sigma} \exp\left[-(t - \mu)^2/(2\sigma^2)\right] dt$, where $\mu$ and $\sigma$ denote the mean and standard deviation. Since this equation lacks an explicit elementary expression, finding a parsimonious equation for approximation based on a small training set is intractable. Nevertheless, our model shows a great capability of approximating the CFD with generalizability (see Appendix Section F for more details).

## 4.3 FREE-FALLING BALLS DATASET

We conducted an experimental evaluation on the free-falling balls dataset to assess the parametric learning capability of our model. The dataset consisted of experimental data of balls dropped from a bridge, as described in de Silva et al. (2020). The dataset comprised 20-30 observations of a ball throw height within the first 2 seconds, aiming to learn the equation governing the ball's

Table 2: MSE of free-falling balls dataset. Details of the equations generated by different models are shown in Appendix E.

| BenchMark | Ours | Ours* | SPL | M-A | M-B | M-C |
|---|---|---|---|---|---|---|
| baseball | **0.053** | 0.068 | 0.300 | 2.798 | 94.589 | 3.507 |
| blue basketball | **0.008** | 0.027 | 0.457 | 0.513 | 69.209 | 2.227 |
| bowling ball | 0.014 | 0.034 | **0.003** | 0.33 | 87.02 | 3.167 |
| golf ball | **0.006** | 0.041 | 0.009 | 0.214 | 86.093 | 1.684 |
| green basketball | 0.094 | **0.045** | 0.088 | 0.1 | 85.435 | 1.604 |
| tennis ball | 0.284 | **0.068** | 0.091 | 0.246 | 72.278 | 0.161 |
| volleyball | 0.033 | **0.025** | 0.111 | 0.574 | 80.965 | 0.76 |
| whiffle ball 1 | **0.038** | 0.660 | 1.58 | 1.619 | 65.426 | 0.21 |
| whiffle ball 2 | **0.041** | 0.068 | 0.099 | 0.628 | 58.533 | 0.966 |
| yellow whiffle ball | 1.277 | 1.080 | **0.428** | 17.341 | 44.984 | 2.57 |
| orange whiffle ball | **0.031** | 0.368 | 0.745 | 0.379 | 36.765 | 3.257 |
| **Average** | **0.173** | 0.242 | 0.356 | 2.24 | 71.02 | 1.828 |

drop and predict the height between 2 and 3 seconds. Since an exact solution for this dataset is not available, we employed the mean squared error (MSE) as our evaluation metric.

We consider two sets of RSRM models, the standard one and the one named RSRM* (denoted by Ours* in Table 2) that fixes the expression form $c_4 x^3 + c_3 x^2 + x_2 x + c_1 + f(x)$. We compared these models with the baseline method SPL (Sun et al., 2023), since other models tend to have large generalization errors due to the limited data points (20-30 per training set) in the falling balls benchmark given the fact that the exact solution is unknown. Three physics models derived from mathematical principles were selected as baseline models for this experiment, and the unknown constant coefficient values were estimated using POWELL (1964). The equations of the baseline models are presented as follows. **M-A**: $h(t) = c_1 t^3 + c_2 t^2 + c_3 t + c_4$, **M-B**: $h(t) = c_1 \exp(c_2 t) + c_3 t + c_4$, and **M-C**: $h(t) = c_1 \log(\cosh(c_2 t)) + c_3$.

The results (see Table 2) show that in most cases, the RSRM model performs better than SPL. The RSRM model can successfully find the equation of motion for uniformly accelerated linear motion $(c_1 x^2 + c_2 x + c_3 + f(x))$ and search for additional terms to minimize the training error. This leads to improved results compared to SPL. However, there are cases where RSRM makes mistakes, such as obtaining expressions in the form of $c_1 \cos(x)^2 + c_2 + f(x)$ when searching for the yellow whiffle ball. This increases the generalization error and reduces the overall effectiveness compared to SPL. Overall, RSRM outperforms SPL in physics equation discovery, demonstrating its effectiveness in solving parametric learning tasks on the free-falling balls dataset.

## 5 ABLATION STUDY

We conducted ablation studies on the Livermore dataset, namely, ablations of the double Q-learning (M-A), the MCTS algorithm (M-B), the MSDB (M-C), the pre-processing step (M-D), and GP (M-E), respectively. We list some of expressions affected by the performance of the model in Table 3.

Table 3: Ablation recovery rate (%) of the Livermore dataset. The specific recovery rates are further shown in Appendix H.

| Equation | Ours | M-A | M-B | M-C | M-D | M-E |
|---|---|---|---|---|---|---|
| $\sin(x_1^2)\cos(x_1) - 2$ | **100** | 100 | 100 | 6 | 100 | 100 |
| $\sin(x_1^3)\cos(x_1^2) - 1$ | **55** | 20 | 0 | 0 | 55 | 0 |
| $\sinh(x_1)$ | **100** | 100 | 100 | 100 | 10 | 100 |
| $\sum_{k=1}^{9} x^k$ | **100** | 83 | 100 | 88 | 100 | 67 |
| $x_1^{1/3}$ | **100** | 100 | 100 | 67 | 100 | 100 |
| $x_1^{2/5}$ | **100** | 100 | 100 | 12 | 100 | 33 |
| **Average:** | **97.95** | 94.36 | 93.64 | 80.45 | 89.45 | 84.95 |

When the double Q-learning module is removed, Model A with only MCTS experiences a decrease in knowledge from previous iterations. This results in reduced search efficiency but increased diversity. As a result, we observe a decrease in performance for equations like $\sum_{k=1}^{9} x^k$, while equations like $\sin(x_1^3)\cos(x_1^2) - 1$ show improved performance. On the other hand, when the MCTS module is removed, Model B with pure double Q-learning tends to overfit more quickly. Consequently, it struggles to produce the most challenging equations, such as $\sin(x_1^3)\cos(x_1^2) - 1$. Similarly, the absence of the expression form search module in Model C limits its ability to discover complex expressions with simple forms, such as $x_1^{1/3}$ and $\sin(x_1^3)\cos(x_1^2) - 1$. Lastly, Model D, without the preprocessing module, suffers a significant reduction in its ability to search for odd and even functions like $\sinh(x_1)$. The removal of the genetic algorithm (M-E) resulted in decreased efficiency across all expression searches. While simpler expressions such as $x_1^{1/3}$ still performed adequately, the performance for complex expressions notably deteriorated.

These observations highlight the importance of all the modules in RSRM. Each module contributes to the overall performance and enables the model to tackle different types of equations effectively.

## 6 CONCLUSION

We have proposed a novel model RSRM that integrates RL techniques, GP, and a modulated sub-tree discovery block to improve the search process for mathematical expressions. Our model outperforms the state-of-the-art baselines in the context of accurately recovering the exact equations for various datasets, and demonstrates superior generalization capabilities. However, one limitation of our current model is the lack of flexibility in setting the expression form as it currently encompasses only three fixed types, which restricts its adaptability to different problem domains. We anticipate future advancements in more flexible methods, e.g., potentially incorporating neural networks to generate slots for SR, utilizing other Q-learning techniques such as prioritized experience replay (Schaul et al., 2015) to enhance the exploitation, etc. Furthermore, we believe that our approach has the potential to be extended to other domains, such as reinforcement learning control tasks. By applying our method to diverse areas, we aim to enhance the performance and applicability of SR techniques.

ACKNOWLEDGMENTS

The work is supported by the National Natural Science Foundation of China (No. 92270118), which is greatly acknowledged. Code and models of Reinforcement Symbolic Regression Machine(RSRM) are available at https://github.com/intell-sci-comput/RSRM.

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

# APPENDIX

## A   MODEL SETTING

In this section, we give more details about the settings of our models.

### A.1   HYPERPARAMETERS

The full set of hyperparameters can be seen in Table S1.

Table S1: Hyperparameters of our model

| Name | Abbreviation | Value |
|------|:---:|:---:|
| **RL parameters** | | |
| Minimum expression lengths | $l_{min}$ | 4 |
| Maximum expression lengths | $l_{max}$ | 35 |
| Maximum number of parameters | $c_{max}$ | 10 |
| Length discount rate | $\eta$ | 0.99 |
| Training rounds | $t_r$ | 50 |
| UCT constant | $c$ | $\sqrt{2}$ |
| Minimum selected times | $n_0$ | 3 |
| Learning rate of double Q-learning | $lr$ | $10^{-3}$ |
| **Genetic Programming parameters** | | |
| GP rounds | $t_{gp}$ | 30 |
| GP population | $p_{gp}$ | 500 |
| GP number of best expressions | $l_b$ | 20 |
| GP Mate rate | $p_{mate}$ | 0.5 |
| GP Mutate rate | $p_{mutate}$ | 0.5 |
| **MSDB parameters** | | |
| Error of Symmetry | $E_{sym}$ | $10^{-5}$ |
| Selection ratio | $k_s$ | 0.1 |
| Expression percentage ratio | $k_p$ | 0.1 |
| Maximum select number | $N$ | 5 |

### A.2   EXPRESSION CONSTRAINT

In this study, we incorporate a prior constraint inspired by the DSR (Petersen et al., 2019) method to effectively reduce the search space for expressions. The following constraints are applied:

- **Length constraint**: The length of expressions is restricted within pre-defined minimum and maximum values. If the current length falls below the minimum threshold, variables ($x$, $y$, $C$, etc.) and parameters will not be generated. Conversely, if the current length, combined with the number of nodes to be generated, reaches the maximum length, only these nodes will be considered.

- **Unary operator constraint**: The direct successor node of a unary operator should not be the inverse of that same operator. This constraint ensures that the generated expressions adhere to the intended structure and prevent redundant combinations.

- **Trigonometric function constraint**: The successor node of a trigonometric function node should not be another trigonometric function. This constraint prevents the generation of expression structures that lead to unnecessary complexity or redundancy.

- **Maximum parameter limit**: A specified maximum number of parameters is imposed to control the complexity of the expressions and prevent overfitting.

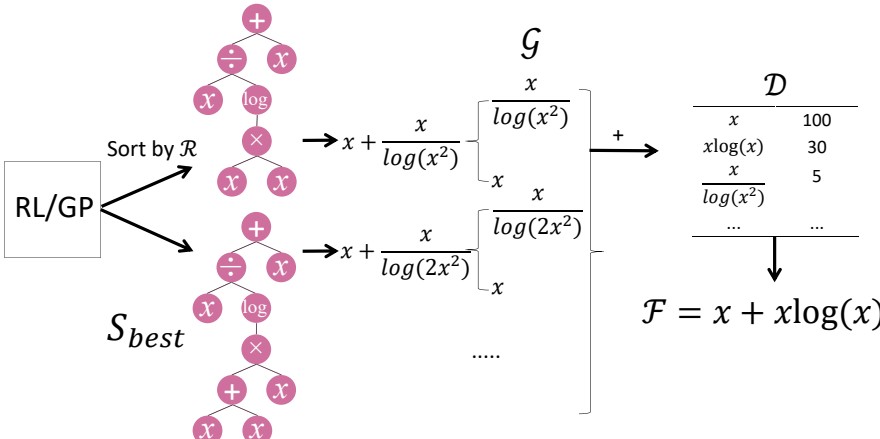

Figure S1: Schematic of the proposed Form-discovery. We first obtain the output of RL/GP and select the equations in which the loss is relatively small. After that, we separate these equations by plus and minus signs and count the number of times these sub-expressions occur overall, and use the smaller equations with more occurrences to cobble together to create new expression forms.

By applying these expression constraints, we aim to enhance the search efficiency and guide the generation of meaningful expressions that align with the desired properties of the target problem.

### A.3 GENETIC PROGRAMMING

Following the generation of expressions by each reinforcement learning algorithm, we engage in the optimization of a predetermined set of expressions. For this purpose, we employ a genetic algorithm, utilizing the DEAP library in Python. This algorithm initializes half of the population using the outcomes of the reinforcement learning process, while the remaining half is generated randomly. Subsequently, we retain the most promising expressions and subject them to further analysis using a subtree analyzer. This process serves to update and refine the expression form, enhancing the overall efficacy of our approach.

### A.4 FORM DISCOVERY

Presented below is an elucidative diagram (refer to Figure S1) pertaining to the process of form discovery as outlined in Algorithm 2. This visual aid serves to elucidate the sequential progression within the algorithm and delineates the roles and interpretations of variables such as $S_{sym}$, $\mathcal{G}$, $\mathcal{D}$, and more.

## B BASELINE MODELS

In this section, we give more details about the settings of baselines.

- **SPL** (Sun et al., 2023): Rooted in the use of MCTS, SPL employs various greedy strategies to ensure efficient exploration. It utilizes full expressions as sub-trees to maximize the use of past information. While excelling in shorter expressions, it falls short in handling longer ones.

- **DSR** (Petersen et al., 2019): DSR takes a gradient-based RL approach along with a recurrent neural network (RNN) that generates a probability distribution over expressions. While effective for longer expressions, it may exhibit limitations in generalization ability.

- **NGGP** (Mundhenk et al., 2021a): Building upon DSR, NGGP enhances its capabilities. Expressions sampled via probability distribution undergo further optimization using GP. The refined expressions then train the RNN with risk-seeking policy gradient.

- **uDSR** (Landajuela et al., 2022): The uDSR amalgamates DSR, AIFeynman, LSPT (Large-scale pre-training), GP, and LM (Linear models). It excels in discovering formulas with constants (favoring polynomial type expressions), but albeit at the cost of increased computation time.

- **DGSR** (Holt et al., 2022): This model leverages pre-trained deep generative models to capture the inherent patterns and structures within equations. This pre-training phase establishes a robust foundation for the subsequent optimization steps conducted through genetic programming.

- **gplearn**(Stephens, 2016): The gplearn offers an efficient and rapid GP-based SR implementation. While proficient in speed, it may exhibit instability and poor scalability.

- **AFP-FE** (Schmidt & Lipson, 2010): Age-Fitness Pareto Optimization is an optimization technique that combines two important factors, age, and fitness, to enhance the performance of evolutionary algorithms. FE means Co-evolved Fitness Predictors.

The full set of hyperparameters can be seen below.

- **SPL**: In line with the original paper, we maintain the same parameter settings for SPL. The discount rate is set to $\eta = 0.9999$, and the candidate operators include addition ($+$), subtraction ($-$), multiplication ($\times$), division ($\div$), cosine ($\cos(\cdot)$), sine ($\sin(\cdot)$), exponential ($\exp(\cdot)$), natural logarithm ($\log(\cdot)$), and square root ($\sqrt{\cdot}$). Other parameter values are as follows: Maximum Module Transplantation: 20, Episodes Between Module Transplantation: 50000, Maximum Tree Size: 50, and Maximum Augmented Grammars: 5.

- **DSR/NGGP/uDSR**: In our study, we adopt the standard parameter configurations as provided in the publicly available implementation of Deep Symbolic Optimization (DSO). This approach entails adjusting two primary hyperparameters. The entropy coefficient is set $\lambda H = 0.05$ and the risk factor is set $\epsilon = 0.005$. Candidate operators are the same as those employed in the SPL. Additionally, NGGP incorporates other hyperparameters related to hybrid methods based on genetic programming. The specific values are listed in Table S2.

- **Genetic Programming (GP)**: We employ the gplearn library for GP-based methods. The hyperparameters for genetic programming are identical to those presented in Table S2.

- **DGSR/AFP-FE**: For both of these models, we exclusively utilized the results obtained from the srbench dataset (La Cava et al., 2021), and the parameters were meticulously tested in accordance with the specifications provided in the official srbench dataset and its associated article.

Table S2: Genetic Programming Hyperparameters on baselines

| Name | Value |
|---|---|
| Rounds | 20 |
| Population | 1000 |
| Mate rate | 0.5 |
| Mutate rate | 0.5 |

## C BASIC BENCHMARK RESULT

### C.1 OVERALL RESULT

In this section, we give more details about the Min Depth and Min Complexity of difficult equations in each benchmark in Table S3.

In this table, the performance of uDSR appears to be subpar, even falling short of its predecessor NGGP. This outcome can be attributed to two primary reasons. Firstly, our application of symbolic learning lacks parameter optimization, except for the Nyugen-c dataset, in which uDSR notably excels due to its compatibility with parameterized scenarios. Secondly, uDSR exhibits a stronger tendency towards generating polynomial functions, whereas

our tests predominantly involve a substantial number of trigonometric and exponential functions. For instance, expressions such as $x_1 x_2 x_3 (\sin(x_4) + \cos(x_5))$ result in complex equations like $x_1 \left[ -0.0437x_1^3 - 5.84x3 - 0.01x4^3 + 0.0038x_4^2 x5 - 0.492x_4 x_5 + x_4(x_2 + \sin(x_5))... \right]$.

## C.2 Nyugen Benchmark Result

Nyugen Benchmark is a standard benchmark for symbolic learning with one or two independent variables and equations randomly sampled over a range. And Nyugen$^c$ is a parametric version of the Nguyen benchmark, allowing the use of parametric optimization to test equations with parameters. In this section, we provide additional details about the results obtained from the Nyugen and Nyugen$^c$ Benchmark experiment.

By referring to Table S4, readers can obtain more detailed information about the performance of each model on each expression, their comparative analysis, and any other relevant insights derived from the experiment.

## C.3 LiverMore Benchmark Result

LiverMore Benchmark contains challenging equations rarely encountered in symbolic learning, including high exponentials, trigonometric functions, and complex polynomials. In this section, we provide additional details about the results obtained from the LiverMore Benchmark experiment.

By referring to Table S5, readers can obtain more detailed information about the performance of each model on each expression, their comparative analysis, and any other relevant insights derived from the experiment.

## C.4 R Benchmark Result

R Benchmark consists of three built-in rational equations with numerous polynomials as divisors and devisees, increasing the learning difficulty. In this section, we provide additional details about the results obtained from the R Rational Benchmark experiment.

By referring to Table S6, readers can obtain more detailed information about the performance of each model on each expression, their comparative analysis, and any other relevant insights derived from the experiment.

Table S3: Minimum Depth of expression tree and Minimum tokens of expression tree as Minimum Complexity of several difficult equations in each benchmark.

| BenchMark | Equation | Min Depth | Min Complexity |
|---|---|---|---|
| Nguyen-5 | $\sin(x_1^2)\cos(x_1) - 1$ | 6 | 12 |
| Nguyen-12 | $x_1^4 - x_1^3 - 0.5x_2^2 + x_2$ | 7 | 24 |
| Nguyen-2$^c$ | $0.48x_1^4 + 3.39x_1^3 + 2.12x_1^2 + 1.78x_1$ | 6 | 25 |
| Nguyen-9$^c$ | $\sin(1.5x_1) + \sin(0.5x_2^2)$ | 5 | 11 |
| LiverMore-3 | $\sin(x_1^3)\cos(x_1^2) - 1$ | 6 | 15 |
| LiverMore-7 | $\sinh(x_1)$ | 6 | 15 |
| LiverMore-16 | $x_1^{2/5}$ | 7 | 17 |
| LiverMore-18 | $\sin(x_1^2)\cos(x_1) - 5$ | 6 | 19 |
| AIFeynman-9 | $x_1 + x_2 + 2\sqrt{x_1 x_2}\cos(x_3)$ | 6 | 17 |
| AIFeynman-10 | $\frac{1}{2}x_1(x_2^2 + x_3^2 + x_4^2)$ | 6 | 20 |
| R-1 | $(x_1 + 1)^3/(x_1^2 - x_1 + 1)$ | 6 | 21 |
| R-2 | $(x_1^5 - 3x_1^3 + 1)/(x_1^2 + 1)$ | 7 | 31 |
| R-3 | $(x_1^5 + x_1^6)/(x_1^4 + x_1^3 + x_1^2 + x_1 + 1)$ | 7 | 35 |

Table S4: Average Recovery Rate (%) of the Nyugen Benchmark over 100 parallel runs

| Name | Equation | Ours | SPL | uDSR | NGGP | DSR | GP |
|------|----------|------|-----|------|------|-----|-----|
| Nguyen-1 | $x_1^3 + x_1^2 + x_1$ | **100** | **100** | **100** | **100** | **100** | 99 |
| Nguyen-2 | $x_1^4 + x_1^3 + x_1^2 + x_1$ | **100** | **100** | **100** | **100** | **100** | 90 |
| Nguyen-3 | $x_1^5 + x_1^4 + x_1^3 + x_1^2 + x_1$ | **100** | **100** | **100** | **100** | **100** | 34 |
| Nguyen-4 | $x_1^6 + x_1^5 + x_1^4 + x_1^3 + x_1^2 + x_1$ | **100** | 99 | **100** | **100** | **100** | 54 |
| Nguyen-5 | $\sin(x_1^2)\cos(x_1) - 1$ | **100** | 95 | 45 | 80 | 72 | 12 |
| Nguyen-6 | $\sin(x_1) + \sin(x_1 + x_1^2)$ | **100** | **100** | **100** | **100** | **100** | 11 |
| Nguyen-7 | $\log(x_1 + 1) + \log(x_1^2 + 1)$ | **100** | **100** | 98 | **100** | 35 | 17 |
| Nguyen-8 | $\sqrt{x_1}$ | **100** | **100** | **100** | **100** | 96 | **100** |
| Nguyen-9 | $\sin(x_1) + \sin(x_2^2)$ | **100** | **100** | 91 | **100** | **100** | 17 |
| Nguyen-10 | $\sin(x_1)\cos(x_2)$ | **100** | **100** | **100** | **100** | **100** | 86 |
| Nguyen-11 | $x_1^{x_2}$ | **100** | **100** | 87 | **100** | **100** | 13 |
| Nguyen-12 | $x_1^4 - x_1^3 - 0.5x_2^2 + x_2$ | **100** | 28 | 30 | 21 | 0 | 0 |
| | Average | **100.00±0.0** | 93.50±11.7 | 87.58±13.6 | 91.75±13.0 | 83.58±18.5 | 44.42±22.1 |
| Nguyen-1[c] | $3.39x_1^3 + 2.12x_1^2 + 1.78x_1$ | **100** | **100** | 58 | **100** | **100** | 0 |
| Nguyen-2[c] | $0.48x_1^4 + 3.39x_1^3 + 2.12x_1^2 + 1.78x_1$ | **100** | 94 | **100** | **100** | **100** | 0 |
| Nguyen-5[c] | $\sin(x_1^2)\cos(x_1) - 0.75$ | **100** | 95 | 67 | 98 | 0 | 1 |
| Nguyen-7[c] | $\log(x_1 + 1.4) + \log(x_1^2 + 1.3)$ | **100** | 0 | **100** | **100** | 93 | 2 |
| Nguyen-8[c] | $\sqrt{1.23x_1}$ | **100** | **100** | **100** | **100** | **100** | 56 |
| Nguyen-9[c] | $\sin(1.5x_1) + \sin(0.5x_2^2)$ | **100** | 98 | 0 | 96 | 0 | 0 |
| Nguyen-10[c] | $\sin(1.5x_1)\cos(0.5x_2)$ | **100** | 0 | 0 | **100** | **100** | 0 |
| | Average | **100.00±0.0** | 69.57±35.2 | 60.71±33.2 | 99.14±1.2 | 70.43±35.7 | 8.43±15.6 |

Table S5: Average Recovery Rate (%) of the LiverMore Benchmark over 100 parallel runs

| Name | Equation | Ours | SPL | uDSR | NGGP | DSR | GP |
|------|----------|------|-----|------|------|-----|-----|
| Livermore-1 | $1/3 + x_1 + \sin(x_1)$ | **100** | 94 | **100** | **100** | 67 | **100** |
| Livermore-2 | $\sin(x_1^2)\cos(x_1) - 2$ | **100** | 29 | 58 | 61 | 26 | 1 |
| Livermore-3 | $\sin(x_1^3)\cos(x_1^2) - 1$ | **55** | 50 | 0 | 2 | 0 | 0 |
| Livermore-4 | $\log(x_1 + 1) + \log(x_1^2 + 1) + \log(x_1)$ | **100** | 61 | **100** | **100** | 72 | **100** |
| Livermore-5 | $x_1^4 - x_1^3 + x_1^2 - x_2$ | **100** | **100** | **100** | **100** | 55 | **100** |
| Livermore-6 | $4x_1^4 + 3x_1^3 + 2x_1^2 + x_1$ | **100** | 8 | **100** | **100** | **100** | **100** |
| Livermore-7 | $\sinh(x_1)$ | **100** | 18 | 0 | 24 | 0 | 0 |
| Livermore-8 | $\cosh(x_1)$ | **100** | 6 | 8 | 30 | 0 | 0 |
| Livermore-9 | $\sum_{i=1}^{9} x_1^i$ | **100** | 21 | **100** | 99 | 18 | 0 |
| Livermore-10 | $6\sin(x_1)\cos(x_2)$ | **100** | 75 | **100** | **100** | 70 | 23 |
| Livermore-11 | $(x_1^2 x_2^2)/(x_1 + x_2)$ | **100** | 0 | **100** | **100** | 78 | 95 |
| Livermore-12 | $x_1^5/x_2^3$ | **100** | **100** | **100** | **100** | 13 | **100** |
| Livermore-13 | $x_1^{1/3}$ | **100** | 12 | **100** | **100** | 59 | 0 |
| Livermore-14 | $x_1^3 + x_1^2 + x_1 + \sin(x_1) + \sin(x_1^2)$ | **100** | **100** | **100** | **100** | 91 | **100** |
| Livermore-15 | $x_1^{1/5}$ | **100** | 0 | **100** | **100** | 28 | 2 |
| Livermore-16 | $x_1^{2/5}$ | **100** | 0 | 60 | 26 | 0 | 0 |
| Livermore-17 | $4\sin(x_1)\cos(x_2)$ | **100** | 89 | **100** | **100** | **100** | 84 |
| Livermore-18 | $\sin(x_1^2)\cos(x_1) - 5$ | **100** | 18 | 59 | 33 | 37 | 0 |
| Livermore-19 | $x_1^5 + x_1^4 + x_1^2 + x_1$ | **100** | 89 | 98 | **100** | **100** | **100** |
| Livermore-20 | $\exp(-x_1^2)$ | **100** | **100** | **100** | **100** | **100** | **100** |
| Livermore-21 | $\sum_{i=1}^{8} x_1^i$ | **100** | 52 | **100** | **100** | 13 | 12 |
| Livermore-22 | $\exp(-0.5x_1^2)$ | **100** | **100** | **100** | **100** | 82 | **100** |
| | Average | **97.95±4.0** | 51.00±16.9 | 81.05±14.6 | 80.68±14.0 | 50.41±15.7 | 50.77±20.4 |

Table S6: Average Recovery Rate (%) of the R Rational Benchmark over 100 parallel runs

| Name | Equation | Ours | SPL | uDSR | NGGP | DSR | GP |
|------|----------|------|-----|------|------|-----|-----|
| $R^0$-1 | $(x_1 + 1)^3/(x_1^2 - x_1 + 1)$ | 5 | 0 | **82** | 15 | 0 | 0 |
| $R^0$-2 | $(x_1^5 - 3x_1^3 + 1)/(x_1^2 + 1)$ | **80** | 0 | 0 | 40 | 0 | 0 |
| $R^0$-3 | $(x_1^5 + x_1^6)/(x_1^4 + x_1^3 + x_1^2 + x_1 + 1)$ | **100** | 0 | 0 | **100** | 0 | 0 |
| $R^*$-1 | $(x_1 + 1)^3/(x_1^2 - x_1 + 1)$ | **48** | 0 | 17 | 2 | 0 | 0 |
| $R^*$-2 | $(x_1^5 - 3x_1^3 + 1)/(x_1^2 + 1)$ | **89** | 0 | 0 | 0 | 0 | 0 |
| $R^*$-3 | $(x_1^5 + x_1^6)/(x_1^4 + x_1^3 + x_1^2 + x_1 + 1)$ | **91** | 0 | 0 | 3 | 0 | 0 |
| | Average | **68.83±28.9** | 0.00±0.0 | 16.50±26.2 | 26.67±31.1 | 0.00±0.0 | 0.00±0.0 |

## C.5 AIFEYNMAN BENCHMARK RESULT

AIFeynman Benchmark contains lots of equations with physical meaning (as part of SRBench (La Cava et al., 2021)), such as expressions for gravity, kinetic energy, and light intensity superposi-

tion. In this section, we provide additional details about the results obtained from the AIFeynman Benchmark experiment.

By referring to Table S7, readers can obtain more detailed information about the performance of each model on each expression, their comparative analysis, and any other relevant insights derived from the experiment.

The selection of 12 AI Feynman equations was based on a stratified representation of difficulty levels. The chosen expressions include easy ones, such as $x_1x_2$ and $\frac{3}{2}x_1x_2$; medium complexity expressions like $x_1x_2x_3\sin(x_4)$ and $x_1x_2 + x_3x_4 + x_5x_6$; medium-hard expressions such as $0.5x_1(x_2^2 + x_3^2 + x_4^2)$ and $x_1x_2x_3(\frac{1}{x_4} - \frac{1}{x_5})$; and hard expressions like $\frac{x_1x_2x_3}{(x_4-x_5)^2+(x_6-x_7)^2+(x_8-x_9)^2}$ and $1 + \frac{x_1x_2}{1-x_1x_2/3}$, providing a comprehensive coverage of the AIFeynman benchmark.

The comprehensive coverage of the AIFeynman benchmark is reflected in Figure 3b of the results section, showcasing the outcomes for all AI Feynman equations considered in the problem set. Notably, our model achieves a state-of-the-art 80% recovery rate, indicating its proficiency in capturing the underlying mathematical structures across a diverse range of expressions.

Table S7: Average Recovery Rate (%) of the AIFeynman Benchmark over 100 parallel runs

| Name | Equation | Ours | SPL | uDSR | NGGP | DSR | GP |
|------|----------|------|-----|------|------|-----|-----|
| AIFeynman-1 | $x_1x_2$ | **100** | 100 | 100 | 100 | 100 | 100 |
| AIFeynman-2 | $\frac{3}{2}x_1x_2$ | **100** | 99 | 97 | 100 | 97 | 87 |
| AIFeynman-3 | $x_1x_2x_3$ | **100** | 100 | 100 | 100 | 100 | 100 |
| AIFeynman-4 | $x_1x_2x_3\sin(x_4)$ | **100** | 98 | 90 | 100 | 100 | 78 |
| AIFeynman-5 | $x_1x_2 + x_3x_4 + x_5x_6$ | **100** | 100 | 100 | 100 | 100 | 82 |
| AIFeynman-6 | $x_1(1 + x_2\cos(x_3))$ | **100** | 100 | 80 | 100 | 100 | 100 |
| AIFeynman-7 | $x_1x_2x_3(\frac{1}{x_4} - \frac{1}{x_5})$ | **100** | 100 | 87 | 100 | 100 | 80 |
| AIFeynman-8 | $x_1(x_2 + x_3x_4\sin(x_5))$ | **100** | 100 | 100 | 100 | 100 | 100 |
| AIFeynman-9 | $x_1 + x_2 + 2\sqrt{x_1x_2}\cos(x_3)$ | **67** | 0 | 8 | 7 | 0 | 0 |
| AIFeynman-10 | $\frac{1}{2}x_1(x_2^2 + x_3^2 + x_4^2)$ | **15** | 0 | 0 | 0 | 0 | 0 |
| AIFeynman-11 | $\frac{x_1x_2x_3}{(x_4-x_5)^2+(x_6-x_7)^2+(x_8-x_9)^2}$ | 0 | 0 | 0 | 0 | 0 | 0 |
| AIFeynman-12 | $1 + \frac{x_1x_2}{1-x_1x_2/3}$ | 0 | 0 | 0 | 0 | 0 | 0 |
| | **Average** | **73.50±24.1** | 66.42±27.8 | 63.50±26.0 | 67.25±27.4 | 66.42±27.8 | 60.58±25.7 |

### C.6 CONST-OPTIMIZATION EXPERIMENT

We conducted an experiment considering different initialization approaches for constant optimization. Specifically, we explored seven methods:

- Case 1. Initializing constants with a vector of ones.
- Case 2. Initializing constants with a vector of random uniform values between 0 and 1.
- Case 3. Initializing constants with a vector of random uniform values between 0.5 and 1.5.
- Case 4. Initializing constants with a vector of random Gaussian values with a mean of 0 and standard deviation of 1.
- Case 5. Initializing constants with a vector of random Gaussian values with a mean of 1 and standard deviation of 1.
- Case 6. Initializing constants with a vector of random Gaussian values with a mean of 1 and standard deviation of 0.5.
- Case 7. Initializing constants with a vector of random Gaussian values with a mean of 0 and standard deviation of $\frac{1}{3}$.

We evaluated the recovery rate of each expression using different constant initialization methods across diverse benchmarks, employing ranges of input values such as [0,1], [-1,1], [0,10], [-10,10], [0,50], [-50,50], and data sizes of 20 or 500. Notably, Table S8 demonstrates that the average recovery rates across all initializing methods are remarkably close.

Examining the distribution of expression recovery rates below 50%, 10%, and 0%, it becomes apparent that the method employing a vector of ones exhibits the highest percentage in Table S9. This observation indicates that the vector of ones initialization method has the highest number of expressions unable to converge across 100 parallel runs.

Table S8: Average Recovery Rate (%) of the Const Optimization Benchmark over 100 parallel runs on each size and each range

| Name | Equation | Case 1 | Case 2 | Case 3 | Case 4 | Case 5 | Case 6 | Case 7 |
|---|---|---|---|---|---|---|---|---|
| Keijzer-1 | $0.3x_1 \sin(6.28x_1)$ | 33.6 | 33.8 | 33.6 | 33.7 | 33.7 | 33.6 | **34.3** |
| Keijzer-4 | $(30.0x_1x_2x_3)/((x_1 - 10.0)x_2^2)$ | **34.2** | 13.3 | 35.7 | 14.8 | 20.7 | 24.3 | 24.1 |
| Keijzer-14 | $8.0/(2.0 + x_1^2 + x_2^2)$ | **100.0** | 99.8 | **100.0** | 82.0 | 95.8 | 99.6 | 99.8 |
| Keijzer-15 | $x_1^{0.6} + x_2^{1.5} - x_2 - x_1$ | 88.0 | 83.2 | **90.3** | 75.5 | 83.8 | 88.3 | 88.7 |
| Nguyen-1[c] | $3.39x_1^3 + 2.12x_1^2 + 1.78x_1$ | 97.2 | **97.5** | **97.5** | 96.8 | 95.9 | 96.9 | 97.4 |
| Nguyen-2[c] | $0.48x_1^4 + 3.39x_1^3 + 2.12x_1^2 + 1.78x_1$ | **98.0** | 92.3 | 97.9 | 92.9 | 95.0 | 97.5 | 96.2 |
| Nguyen-5[c] | $\sin(x_1^2)\cos(x_1) - 0.75$ | 100.0 | 100.0 | 100.0 | 100.0 | 100.0 | 100.0 | 100.0 |
| Nguyen-7[c] | $\log(x_1 + 1.4) + \log(x_1^2 + 1.3)$ | 100.0 | 53.4 | 78.4 | 50.6 | 56.0 | 70.4 | 97.1 |
| Nguyen-8[c] | $\sqrt{1.23x_1}$ | 100.0 | 100.0 | 100.0 | 99.5 | 100.0 | 100.0 | 100.0 |
| Nguyen-9[c] | $\sin(1.5x_1) + \sin(0.5x_2^2)$ | 35.3 | **47.9** | 43.4 | 42.6 | 43.8 | 45.6 | 46.3 |
| Nguyen-10[c] | $\sin(1.5x_1)\cos(0.5x_2)$ | 35.1 | **47.7** | 38.7 | 42.8 | 42.7 | 44.3 | 42.3 |
| Nguyen-11[c] | $2.7x_1^{x_2}$ | 99.2 | 88.2 | 98.5 | 79.3 | 89.3 | 95.8 | **100.0** |
| Jin[*]-1 | $2.5x_1^4 - 1.3x_1^3 + 0.5x_1^2 - 1.7x_1$ | **98.3** | 97.5 | 96.3 | 97.0 | 94.7 | 97.7 | 97.4 |
| Jin[*]-2 | $8.0x_1^3 - 8.0x_1^2 + 15.0x_1$ | 81.0 | 83.0 | 81.4 | 83.2 | 83.1 | 82.9 | 82.5 |
| Jin[*]-3 | $0.7x_1^3 - 1.7x_1$ | 100.0 | 54.1 | 92.2 | 52.9 | 76.1 | 83.3 | 87.9 |
| Jin[*]-4 | $1.5\exp(x_1) + 5.0\cos(x_1)$ | 87.7 | 93.8 | **94.6** | 88.1 | 92.3 | 94.5 | 93.6 |
| Jin[*]-5 | $6.0\sin(x_1)\cos(x_1)$ | 100.0 | 100.0 | 100.0 | 100.0 | 100.0 | 100.0 | 100.0 |
| Jin[*]-6 | $1.35x_1^2 + 5.5\sin((x_1 - 1)^2)$ | 100.0 | 100.0 | 100.0 | 100.0 | 100.0 | 100.0 | 100.0 |
| **Average** | | 81.06 | 76.19 | 80.90 | 73.46 | 77.17 | 79.74 | **81.08** |

Table S9: Persentage (%) of expressions under certain recovery rate about the Const Optimization Benchmark over each initializing method

| Recovery Rate | Case 1 | Case 2 | Case 3 | Case 4 | Case 5 | Case 6 | Case 7 |
|---|---|---|---|---|---|---|---|
| **<50%** | 18.04 | 25.26 | 18.04 | 26.29 | 19.07 | 17.53 | **17.01** |
| **<10%** | 15.98 | **11.34** | 13.92 | 11.86 | 11.86 | **11.34** | 12.37 |
| **0%** | 7.22 | 4.12 | 6.19 | 3.61 | 4.64 | 3.61 | **3.09** |

## C.7 TRADE-OFF EXPERIMENT

We used five different configurations for symbolic regression on the Nyugen dataset and obtained five different sets of results. We calculated the curves of the average number of tests about the expression recovery rate for all data/ Nyugen-4/ Nyugen-5/ Nyugen-11/ Nyugen-12 for the five different data with different configurations as shown in Figure S2.

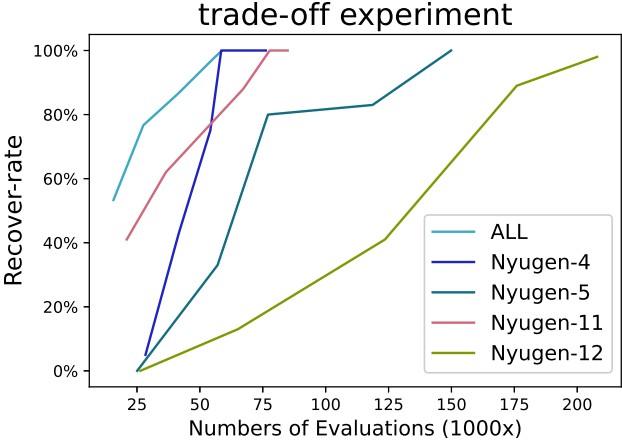

Figure S2: Trade-off between accuracy and number of evaluations in Nyugen Benchmark.

The evaluation count of the RMSM exhibits a discernible pattern, roughly falling within two ranges: the first encompasses around 80% of the recovery rate, while the second spans from 80% to 100%.

Table S10: Trade-off experiment: Average Evaluation Number /Average Recovery Rate (%) of the Nyugen Benchmark over 100 parallel runs

| Name | 5 epochs | 15 epochs | 25 epochs | 35 epochs | 45 epochs |
|---|---|---|---|---|---|
| ALL | 15654/53 | 27543/77 | 41525/87 | 55395/98 | 58240/100 |
| Nyugen-4 | 28389/5 | 41638/43 | 54151/75 | 58483/100 | 76216/100 |
| Nyugen-5 | 25084/0 | 57033/33 | 77075/80 | 118661/83 | 149868/100 |
| Nyugen-11 | 20932/41 | 36487/62 | 67237/88 | 77757/100 | 84902/100 |
| Nyugen-12 | 26345/0 | 65122/13 | 123571/41 | 175970/89 | 207889/98 |

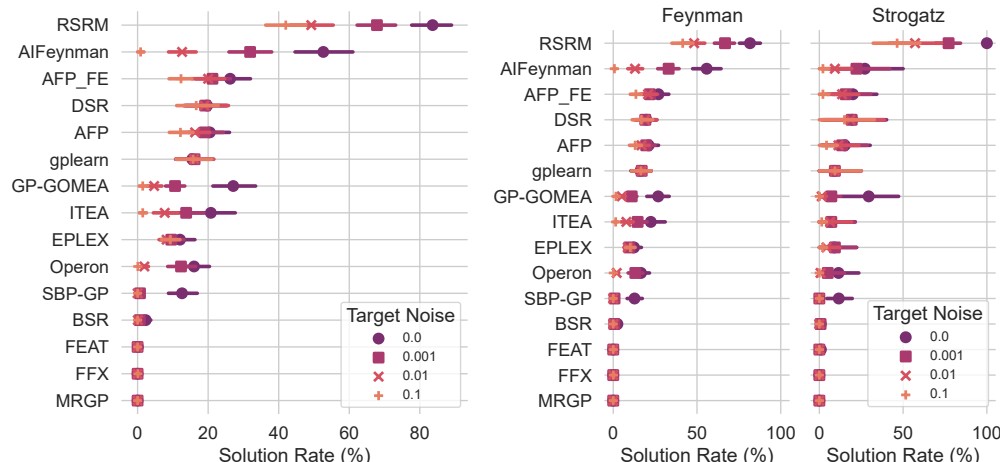

Figure S3: Result of SRBench with 10 parallel runs for each dataset.

Both ranges can be approximated with linear functions; however, the second range displays a notably steeper slope, indicating that achieving higher recovery rates beyond 80% becomes considerably more challenging.

We categorized RSRM into using 5, 15, 25, 35, 45 epochs and tested the average number of tests and the average recovery rate for each different environment and different data respectively in Table S10.

# D SRBENCH RESULT

We used the full set of SRBench for testing, and the overall results are as follows in Figure S3, including both the AIFeynman dataset and the Strogatz dataset.

# E FREE-FALLING BALLS DATASET RESULT

In this section, we provide additional details about the results obtained from the Falling-Balls dataset experiment. To improve the performance of the SPL model, we incorporated the operators $\log(\cosh(\cdot))$. This addition aimed to enhance the model's ability to capture the underlying patterns in the data.

The complete results of the experiment, including the functions found, can be found in Table S11.

Table S11: Functions generated in Falling-Balls Experiment

| Name | Model | Equation |
|---|---|---|
| baseball | Ours | $-4.43t^2 + 0.36\sin(t^2 + 1.51)^2 + 47.35$ |
| | Model A | $0.09t^3 - 5.47t^2 + 2.47t + 46.52 + \cos(t^2 - 2.5t)^{0.5}$ |
| | SPL | $-4.54t^2 + 0.625t + 47.8$ |
| blue basket ball | Ours | $-1.66t^3 - 4.95t^2\cos(\sqrt{t}) + 46.46$ |
| | Model A | $-0.1t^3 - 4.49t^2 + 37.54t + 46.49 - t(\cos(t) + 36.77)$ |
| | SPL | $-0.25t^4 + t^3 - 5.11t^2 + 46.47$ |
| bowling ball | Ours | $-4.63t^2 + \sin(0.83t)\sin(t) + 46.13$ |
| | Model A | $0.18t^3 - 6.0t^2 + 2.15t + 45.43 + |t - 0.62|$ |
| | SPL | $-0.285t^3 - 3.82t^2 + 4.14 \times 10^{-5}\exp(20.74t^2 - 12.45t^3) + 46.1$ |
| golf ball | Ours | $-0.09t^3 - 4.44t^2 + 5.26 \times 10^{-5}t/\log(t) + 49.51$ |
| | Model A | $-2.18t^3 + 11.75t^2 + 1.96t + 25.86 - 2.36\exp(t) + 25.98\cos(t)$ |
| | SPL | $-4.9633t^2 + \log(\cosh(t)) + 49.5087$ |
| green basket ball | Ours | $46.34 - 4.15t^2$ |
| | Model A | $-0.09t^3 - 4.59t^2 + 1.6t + 45.26 + (\frac{0.02\sqrt{t}}{t - \exp(\cos(t))} - t + 1)\cos(t)$ |
| | SPL | $-4.1465t^2 + 45.9087 + \log(\cosh(1))$ |
| tennis ball | Ours | $47.78\cos(0.43t - 0.02)$ |
| | Model A | $0.33t^3 - 4.9t^2 + 0.66t + 47.74$ |
| | SPL | $-4.0574t^2 + \log(\cosh(0.121t^3)) + 47.8577$ |
| volleyball | Ours | $48.15 - 3.67(t + 0.03)^2$ |
| | Model A | $1.59t^3 - 11.1t^2 + 0.93t + 58.53 - 10.53\cos(t)$ |
| | SPL | $-3.78t^2 + 48.0744$ |
| whiffle ball1 | Ours | $-t^2(3.83 - 0.31t) + 47.07$ |
| | Model A | $-0.08t^3 - 2.17t^2 - 1.69t + 46.29 + \sqrt{t + \sin(3t)}$ |
| | SPL | $-t^3 + 4.16t^2 + 47.01\exp(-0.15t^2)$ |
| whiffle ball2 | Ours | $-2.18t^2 + 0.1t\cos(t) + 3.35\cos(t) + 43.88$ |
| | Model A | $0.46t^3 - 4.39t^2 + 0.19t + 47.26 - 0.05\cos(\exp(t))$ |
| | SPL | $65.86\exp(-0.0577t^2) - 18.61$ |
| yellow whiffle ball | Ours | $(\cos(1.75t) + 47.59)\cos(0.36t)$ |
| | Model A | $-0.27t^3 - 2.58t^2 - 2.5t + 48.25 + (t + 0.41)\exp(\sqrt{t + t^2 - 2\sqrt{t^3}})$ |
| | SPL | $(148.99 - 14.58t^2 + 48.96\log(\cosh(x)))/(\log(\cosh(t)) + 3.065)$ |
| orange whiffle ball | Ours | $-17.82t - 33.11/\exp(t)^{0.5} + 80.94$ |
| | Model A | $0.42t^3 - 3.81t^2 - 1.4t + 47.84$ |
| | SPL | $-1.66t + 47.86\exp(-0.0682t^2)$ |

## F    GENERALIZATION EXPERIMENT RESULT

To compare the generalization ability of our model with other methods, we conducted an experiment on generalization performance. The dataset was generated using the cumulative distribution function (CDF) defined as

$$F(x, \mu, \sigma) = \int_{-\infty}^{x} \frac{1}{\sqrt{2\pi}\sigma} e^{-\frac{(t - \mu)^2}{2\sigma^2}} dt$$

with varying means ($\mu$) and variances ($\sigma$). The dataset consisted of 201 points spanning the range from $-100$ to 100. Each dataset is divided into three subsets: a training set, a test set, and a validation set. The training set comprises points ranging from 30 to 80, while the test set consists of points ranging from 10 to 25. The validation set covers a broader range, spanning from 0 to 100.

Because this equation lacks an explicit elementary expression, it's impossible to obtain an analytical solution for the entire curve by reducing the training error to zero. Therefore, learning this function

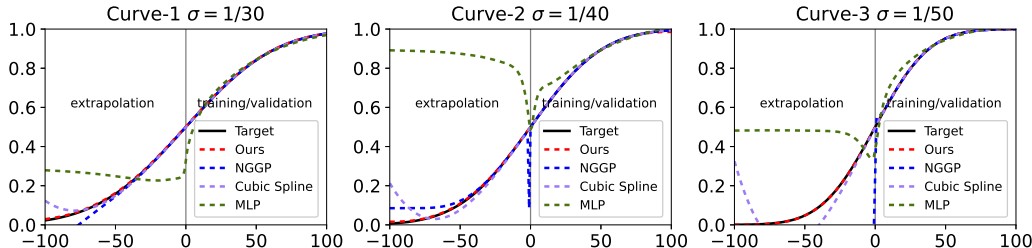

Figure S4: Result of generalization test experiment.

requires balancing between training error and generalization error, demanding a stronger ability to generalize. Moreover, only half of the curve data is provided, necessitating strong generalization skills from the model to extrapolate and accurately fit the missing portion of the curve. This calls for the proficiency of the model in capturing the distribution of the entire curve.

To evaluate the performance of our approach and compare it with baselines, we define the following settings for each method:

- **Ours**: The training set is utilized for generating expressions and calculating the corresponding rewards. The test set is employed to evaluate the quality of the generated expressions. Finally, the validation set is employed to select the most promising expressions from the outputs.

- **NGGP**(Mundhenk et al., 2021a): Both the training set and the test set are used for generating expressions and computing rewards. The HallOfFame, which contains the best expressions, is then leveraged to choose expressions using the validation set.

- **Linear regression**: The training set and the test set are employed for training the linear regression model.

- **Cubic splines**: The training set and the test set are used to train the cubic spline model.

- **Deep learning**: In the deep learning approach, we employ a Multilayer Perceptron (MLP) architecture with one input, one output, and a hidden layer ranging in size from 30 to 50. We set the learning rate to $10^{-3}$ and train 100 epochs. We experiment with different configurations of the hidden layer and select the model that yields the best performance. The MLP is trained using the training set, and the test set is used to evaluate the performance of each model. By varying the size of the hidden layer, we aim to find the optimal architecture that achieves the highest accuracy or lowest error on the given task.

The full results of the generalization experiment can be found in Table S12 and Figure S4. This table presents a detailed overview of the performance of the model and the baselines. Additionally, Table S13 presents the equations discovered by the model and the baselines. These tables demonstrate that our model outperforms the baseline methods in terms of generalization ability. The curves fitted by our model exhibit better accuracy and capture the underlying patterns in the data more effectively.

## G    PARITY DETERMINATION EXPERIMENT

In this section, we conduct an experiment to compare the efficiency and effectiveness of form discovery by AIFeynman(Udrescu & Tegmark, 2020) and our method.

According to AIFeynman (Udrescu & Tegmark, 2020), it is also possible to use MLP as a learning curve to determine the parity of a function, so we compared the efficiency and effectiveness of MLP and cubic splines in determining parity.

The configuration of the MLP follows the structure employed in AIFeynman. It consists of a five-layer neural network with a balanced training set and validation set ratio of 5:5. The input layer accepts one variable and yields an output of 128. Subsequent to the input layer, the hidden layers encompass input-output feature pairs of 128-128, 128-64, and 64-64. The output layer produces a single variable. Optimization is executed using the Adam optimizer, and the activation function for each layer is set

Table S12: Mean Squared Error (MSE) of each method and each part of the curve in the Generalization Experiment

| Name | Ours | NGGP | Linear | Cubic Splines | MLP |
|---|---|---|---|---|---|
| total error on curve 1 | $\mathbf{1.05 \times 10^{-5}}$ | 0.00215 | 0.0114 | 0.000381 | 0.0142 |
| total error on curve 2 | $\mathbf{9.79 \times 10^{-6}}$ | 0.00163 | 0.0297 | 0.00162 | 0.261 |
| total error on curve 3 | $\mathbf{2.61 \times 10^{-7}}$ | 0.327 | 0.0821 | 0.00563 | 0.0762 |
| extrapolation error on curve 1 | $\mathbf{1.65 \times 10^{-5}}$ | 0.00429 | 0.0215 | 0.000758 | 0.0278 |
| extrapolation error on curve 2 | $\mathbf{1.41 \times 10^{-5}}$ | 0.00324 | 0.0565 | 0.00323 | 0.518 |
| extrapolation error on curve 3 | $\mathbf{2.67 \times 10^{-7}}$ | 0.65 | 0.158 | 0.0112 | 0.151 |
| validation error on curve 1 | $4.46 \times 10^{-6}$ | $\mathbf{2.59 \times 10^{-10}}$ | 0.00129 | $2.76 \times 10^{-10}$ | 0.000532 |
| validation error on curve 2 | $5.45 \times 10^{-6}$ | $\mathbf{1.08 \times 10^{-10}}$ | 0.00265 | $2.87 \times 10^{-9}$ | 0.00177 |
| validation error on curve 3 | $2.55 \times 10^{-7}$ | $2.94 \times 10^{-5}$ | 0.00518 | $\mathbf{3.75 \times 10^{-8}}$ | 0.00106 |
| test error on curve 1 | $6.95 \times 10^{-6}$ | $9.09 \times 10^{-12}$ | 0.000545 | $\mathbf{< 1 \times 10^{-12}}$ | 0.000253 |
| test error on curve 2 | $2.25 \times 10^{-7}$ | $\mathbf{< 1 \times 10^{-12}}$ | 0.00127 | $\mathbf{< 1 \times 10^{-12}}$ | 0.00344 |
| test error on curve 3 | $8.8 \times 10^{-8}$ | $\mathbf{< 1 \times 10^{-12}}$ | 0.00272 | $\mathbf{< 1 \times 10^{-12}}$ | 0.00358 |
| training error on curve 1 | $3.88 \times 10^{-6}$ | $3.57 \times 10^{-12}$ | 0.000254 | $\mathbf{< 1 \times 10^{-12}}$ | $6.93 \times 10^{-5}$ |
| training error on curve 2 | $1.26 \times 10^{-6}$ | $\mathbf{< 1 \times 10^{-12}}$ | 0.000524 | $\mathbf{< 1 \times 10^{-12}}$ | $8.6 \times 10^{-5}$ |
| training error on curve 3 | $3.34 \times 10^{-7}$ | $7.14 \times 10^{-12}$ | 0.000905 | $\mathbf{< 1 \times 10^{-12}}$ | $7.9 \times 10^{-5}$ |

Table S13: Functions generated in Generalization Experiment. Our functions are easier to calculate and shorter than NGGP's.

| Name | Model | Equation |
|---|---|---|
| curve-1 | Ours | $0.503 + (117.088x)/(x^2 + 14702)$ |
| | NGGP | $\cos(\exp((0.49x \log(0.028x + 29.5/(0.039x + 9.82)) - 2.78)/(0.115x - 60.5)))$ |
| curve-2 | Ours | $(6.08x + 0.785)/(0.0639x^2 + 615.179) + 0.50003$ |
| | NGGP | $\cos(2.95 \exp(-0.68 \exp(0.41 \exp(5.5x \exp(24.67/(115.6 \exp((4.75x + 13.3)/x) + 3.2))/(2x + 214.3)))))$ |
| curve-3 | Ours | $(x \sin(371.57/(13928/x + x)) + x)/(0.0024 + 2x)$ |
| | NGGP | $\cos(\log(1 + 1.67 \exp(-1.56/(\exp(17.7 \exp(\exp((-12.9 + 4.95 \log(x)/x)/x))/x) - 1.16 + 0.656/x))))$ |

to the hyperbolic tangent (tanh). The training process encompasses 2000 rounds, initiated with a learning rate of 0.01, which is subsequently reduced by a factor of 0.1 whenever the loss increases.

We used seven functions, the top three are odd functions, from easy to hard, then the next three are even functions, and the last one is a non-odd non-even function.

We first tested the speed of both methods. The cubic spline method takes 0.216, 0.217, 0.219, 0.220, 0.226 milliseconds to run on 10, 100, 1000, 10000, and 100000 data points, respectively. Correspondingly, MLP takes 3.28, 3.64, 8.97, 64.1, 549.7 seconds to run.

We then tested two loss equations for the corresponding functions, as shown in the Table S14, and the loss functions are as follows: $L_{odd} = \sum_{i=1}^{n}(y(x) + y(-x))^2/n$, $L_{even} = \sum_{i=1}^{n}(y(x) - y(-x))^2/n$. It can be seen in Figure S5 that if $10^{-4}$ of MSE is used as the cutoff for whether it is an odd/even function or not, the amount of data required by MLP is about 100–1000 times more than that of the spline.

# H   ABLATION EXPERIMENT RESULT

In this section, we provide more detailed information about the results obtained from the ablation experiment on LiverMore benchmark. The full results of the ablation experiment can be found in Table S15. This table presents a comprehensive overview of the performance of the model under different ablation settings.

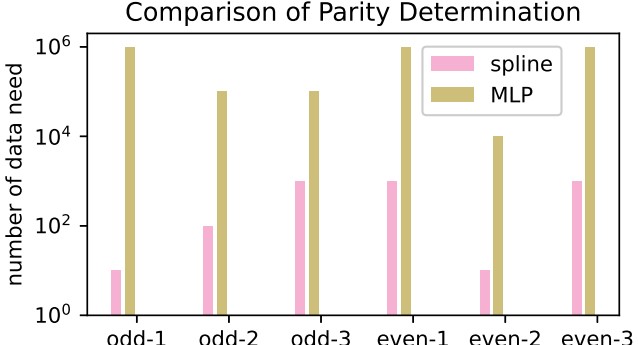

Figure S5: The parity determination performance test. We test the ability of parity determination of two models by three odd functions and even functions from easy to hard. Compete Setting of this experiment is shown in Table S14.

Table S14: The average loss results for the Parity Determination Experiment across 10 parallel runs are presented in the table below. Each cell in the table contains two values: the upper value, situated above the horizontal line, signifies the loss value of the MLP, while the lower value indicates the loss of the cubic splines.

| Name | Equation | Input Range | 10 | 100 | 1000 | 10000 | 100000 |
|---|---|---|---|---|---|---|---|
| odd-1 | $x$ | $[-1,1]$ | $\frac{2.84\times10^{-1}}{1.99\times10^{-16}}$ | $\frac{2.57\times10^{-3}}{5.58\times10^{-17}}$ | $\frac{1.15\times10^{-3}}{2.78\times10^{-17}}$ | $\frac{5.88\times10^{-4}}{3.47\times10^{-18}}$ | $\frac{1.21\times10^{-4}}{1.35\times10^{-20}}$ |
| | | $[-5,5]$ | $\frac{3.04\times10^{-1}}{1.40\times10^{-15}}$ | $\frac{3.31\times10^{-3}}{3.18\times10^{-17}}$ | $\frac{3.37\times10^{-3}}{0.00\times10^{0}}$ | $\frac{9.79\times10^{-4}}{1.55\times10^{-17}}$ | $\frac{1.22\times10^{-4}}{4.85\times10^{-19}}$ |
| odd-2 | $x+\sinh(x)+x^3$ | $[-1,1]$ | $\frac{2.15\times10^{-1}}{5.59\times10^{-4}}$ | $\frac{2.04\times10^{-3}}{7.02\times10^{-7}}$ | $\frac{1.58\times10^{-3}}{1.77\times10^{-9}}$ | $\frac{1.40\times10^{-4}}{4.51\times10^{-13}}$ | $\frac{1.11\times10^{-4}}{2.16\times10^{-12}}$ |
| | | $[-5,5]$ | $\frac{7.70\times10^{-1}}{2.85\times10^{0}}$ | $\frac{5.37\times10^{-2}}{2.05\times10^{-3}}$ | $\frac{4.53\times10^{-2}}{5.22\times10^{-7}}$ | $\frac{1.17\times10^{-3}}{2.70\times10^{-10}}$ | $\frac{3.96\times10^{-4}}{1.99\times10^{-12}}$ |
| odd-3 | $x^3+x+x^5+$ $\sin(x)\times\cosh(x)$ | $[-1,1]$ | $\frac{1.64\times10^{-1}}{3.10\times10^{-2}}$ | $\frac{1.09\times10^{-3}}{8.76\times10^{-5}}$ | $\frac{5.76\times10^{-4}}{2.52\times10^{-8}}$ | $\frac{2.38\times10^{-4}}{8.57\times10^{-12}}$ | $\frac{8.04\times10^{-5}}{4.00\times10^{-11}}$ |
| | | $[-5,5]$ | $\frac{7.60\times10^{-2}}{3.45\times10^{2}}$ | $\frac{7.07\times10^{-4}}{8.16\times10^{-2}}$ | $\frac{6.28\times10^{-5}}{6.28\times10^{-5}}$ | $\frac{1.55\times10^{-4}}{2.75\times10^{-8}}$ | $\frac{6.95\times10^{-5}}{1.62\times10^{-9}}$ |
| even-1 | $x^2$ | $[-1,1]$ | $\frac{2.60\times10^{-1}}{2.64\times10^{-14}}$ | $\frac{4.33\times10^{-3}}{5.11\times10^{-15}}$ | $\frac{7.44\times10^{-4}}{3.01\times10^{-15}}$ | $\frac{1.59\times10^{-4}}{1.78\times10^{-13}}$ | $\frac{8.76\times10^{-5}}{7.81\times10^{-13}}$ |
| | | $[-5,5]$ | $\frac{1.85\times10^{-1}}{1.33\times10^{-13}}$ | $\frac{2.13\times10^{-3}}{2.73\times10^{-13}}$ | $\frac{6.02\times10^{-4}}{1.38\times10^{-13}}$ | $\frac{3.53\times10^{-4}}{1.20\times10^{-12}}$ | $\frac{2.22\times10^{-4}}{1.95\times10^{-11}}$ |
| even-2 | $x\times\sinh(x)$ | $[-1,1]$ | $\frac{7.25\times10^{-2}}{2.43\times10^{-4}}$ | $\frac{1.20\times10^{-2}}{2.93\times10^{-7}}$ | $\frac{9.35\times10^{-4}}{5.77\times10^{-10}}$ | $\frac{2.65\times10^{-4}}{3.23\times10^{-13}}$ | $\frac{7.32\times10^{-5}}{2.70\times10^{-12}}$ |
| | | $[-5,5]$ | $\frac{9.94\times10^{-1}}{5.54\times10^{0}}$ | $\frac{7.90\times10^{-1}}{1.58\times10^{-3}}$ | $\frac{8.06\times10^{-1}}{1.55\times10^{-5}}$ | $\frac{2.88\times10^{-1}}{1.00\times10^{-9}}$ | $\frac{3.15\times10^{-4}}{2.02\times10^{-10}}$ |
| even-3 | $x^4+\log(x^2+1)+$ $\cos(x)\times\exp(0.1x^2)$ | $[-1,1]$ | $\frac{1.09\times10^{-1}}{3.68\times10^{-2}}$ | $\frac{2.61\times10^{-3}}{7.85\times10^{-6}}$ | $\frac{1.52\times10^{-3}}{1.53\times10^{-8}}$ | $\frac{3.09\times10^{-4}}{3.20\times10^{-12}}$ | $\frac{1.48\times10^{-4}}{6.55\times10^{-12}}$ |
| | | $[-5,5]$ | $\frac{9.49\times10^{-1}}{5.20\times10^{0}}$ | $\frac{9.30\times10^{-1}}{3.61\times10^{-2}}$ | $\frac{9.21\times10^{-1}}{2.10\times10^{-5}}$ | $\frac{6.65\times10^{-1}}{2.54\times10^{-9}}$ | $\frac{7.76\times10^{-4}}{3.09\times10^{-10}}$ |
| none | $x^3+\log(x^2+1)+$ $x^7+\sinh(x)$ | $[-1,1]$ | $\frac{9.90\times10^{-1}}{1.68\times10^{0}}$ | $\frac{4.99\times10^{-1}}{7.05\times10^{0}}$ | $\frac{5.13\times10^{-1}}{2.23\times10^{1}}$ | $\frac{5.13\times10^{-1}}{6.81\times10^{1}}$ | $\frac{5.16\times10^{-1}}{2.16\times10^{2}}$ |
| | | $[-5,5]$ | $\frac{9.11\times10^{-1}}{6.95\times10^{2}}$ | $\frac{9.16\times10^{-1}}{4.22\times10^{1}}$ | $\frac{9.22\times10^{-1}}{1.30\times10^{2}}$ | $\frac{6.81\times10^{-1}}{4.16\times10^{2}}$ | $\frac{4.20\times10^{-3}}{1.31\times10^{3}}$ |

Table S15: Average Recovery Rate (%) of the Ablation Experiment over 100 parallel runs

| Name | Equation | Ours | ModelA | ModelB | ModelC | ModelD |
|---|---|---|---|---|---|---|
| Livermore-1 | $1/3 + x_1 + \sin(x_1)$ | **100** | **100** | **100** | **100** | **100** |
| Livermore-2 | $\sin(x_1^2)\cos(x_1) - 2$ | **100** | **100** | **100** | 6 | **100** |
| Livermore-3 | $\sin(x_1^3)\cos(x_1^2) - 1$ | **55** | 20 | 0 | 0 | **55** |
| Livermore-4 | $\log(x_1 + 1) + \log(x_1^2 + x_1) + \log(x_1)$ | **100** | **100** | **100** | **100** | **100** |
| Livermore-5 | $x_1^4 - x_1^3 + x_1^2 - x_2$ | **100** | **100** | **100** | **100** | **100** |
| Livermore-6 | $4x_1^4 + 3x_1^3 + 2x_1^2 + x_1$ | **100** | **100** | **100** | **100** | **100** |
| Livermore-7 | $\sinh(x_1)$ | **100** | **100** | **100** | **100** | 10 |
| Livermore-8 | $\cosh(x_1)$ | **100** | **100** | **100** | **100** | 3 |
| Livermore-9 | $\sum_{i=1}^{9} x_1^i$ | **100** | 83 | **100** | 88 | **100** |
| Livermore-10 | $6\sin(x_1)\cos(x_2)$ | **100** | **100** | **100** | **100** | **100** |
| Livermore-11 | $(x_1^2 x_2^2)/(x_1 + x_2)$ | **100** | 91 | **100** | **100** | **100** |
| Livermore-12 | $x_1^5/x_2^3$ | **100** | **100** | **100** | **100** | **100** |
| Livermore-13 | $x_1^{1/3}$ | **100** | **100** | **100** | 67 | **100** |
| Livermore-14 | $x_1^3 + x_1^2 + x_1 + \sin(x_1) + \sin(x_1^2)$ | **100** | **100** | **100** | **100** | **100** |
| Livermore-15 | $x_1^{1/5}$ | **100** | **100** | **100** | 97 | **100** |
| Livermore-16 | $x_1^{2/5}$ | **100** | **100** | **100** | 12 | **100** |
| Livermore-17 | $4\sin(x_1)\cos(x_2)$ | **100** | **100** | **100** | **100** | **100** |
| Livermore-18 | $\sin(x_1^2)\cos(x_1) - 5$ | **100** | 89 | 90 | 0 | **100** |
| Livermore-19 | $x_1^5 + x_1^4 + x_1^2 + x_1$ | **100** | **100** | **100** | **100** | **100** |
| Livermore-20 | $\exp(-x_1^2)$ | **100** | **100** | **100** | **100** | **100** |
| Livermore-21 | $\sum_{i=1}^{8} x_1^i$ | **100** | **100** | **100** | **100** | **100** |
| Livermore-22 | $\exp(-0.5x_1^2)$ | **100** | **100** | **100** | **100** | **100** |
| | **Average** | **97.95±4.0** | 94.68±7.2 | 95.00±8.9 | 80.45±15.6 | 89.45±11.9 |

