# OpenReview forum: "Reinforcement Symbolic Regression Machine"
_ICLR.cc/2024/Conference — ICLR 2024 poster_

### Official Review · Reviewer_XG8a · 2023-10-20

**Soundness:** 3 good
**Presentation:** 2 fair
**Contribution:** 2 fair
**Rating:** 6
**Confidence:** 4

**Summary:**

The paper proposes a novel Reinforcement Symbolic Regression Machine (RSRM) method for symbolic regression. The method is based on Monte Carlo tree search (MCTS) and double Q-learning. The method is evaluated on several benchmark datasets and compared with several representative baseline models. The paper contains a detailed ablation study to demonstrate the effectiveness of each component of the proposed method.

**Strengths:**

- The paper contains extensive experiments on a variety of benchmark datasets, demonstrating the superiority of the proposed method over several representative baseline models.
- The paper contains a detailed ablation study to demonstrate the effectiveness of each component of the proposed method.

**Weaknesses:**

- The paper has space for improvement in terms of presentation. The paper is not well written and the authors should carefully proofread the paper to improve the presentation quality.

**Questions:**

- (Abstract) "In nature, the behaviors". Consider changing to "In nature, the behavior"
- (Abstract) "Automatically distilling these equations from limited data is cast as a symbolic regression (SR)". This sentence is confusing. Symbolic Regression **is** the task of distilling equations from data. The sentence gives the impression that SR is something else and that the authors are using SR to solve that something else. Please clarify.
- (Abstract) "The discrete search space tends toward infinity". This statement is not mathematically rigorous. The search space in SR (with variable length expressions) is always infinite. Do the authors mean the complexity of the search space? How do they measure it? Please clarify.
- (Introduction) "The early process of fitting expressions involves". What do the authors mean by "early process"? Are they taking a historical perspective of the field? If so, please clarify.
- (Introduction) " By incorporating double Q- learning into MCTS, we effectively address issues of overfitting and generate new expressions". How does Q-learning address overfitting? Please clarify. What do the authors mean with "generate new expressions"? Any other SR method is "generating new expressions" as well.
- (Method) " we introduce an interpolation method (e.g., data pre-processing)". What do the authors mean by "interpolation method" and how is "data pre-processing" related to it? Please clarify.
- (Method) "the MSDB examines whether the few expressions that perform well adhere to a specific form. For instance, if both $\exp(x) − x$ and $\exp(x) + x$ yield favorable results, the expression can be confirmed as $\exp(x) − f(x)$, thereby allowing us to focus on finding $f(x)$" How do the authors determine what "perform well" means? I can understand the intuition of the authors: if both $\exp(x) − x$ and $\exp(x) + x$ yield favorable results, then the expression must be close to the middle of both expressions, i.e., $\exp(x)$. However, it seems to me that how to determine the threshold in the fitting error to determine what "perform well" means is not clear. Is this threshold the same for all datasets? If not how is it determined? Please clarify.
- (Method) "This approach effectively reduces the difficulty associated with specific equations." This statement is not clear. Please clarify.
- (Expression Tree) "The underlying objective of SR is to transform the given task into the generation of an optimal expression tree". This statement is not clear. Which task is SR transforming into the generation of an optimal expression tree? Please clarify.
- (Modulated sub-tree discovery) "This search form focuses on identifying expressions of the form like ex − x and ex + x." What do the authors mean by "search form"? Please clarify.
- (Modulated sub-tree discovery) "In this search form, we obtain good expressions such as 1.57ex and 1.56ex + x,". The concept of "search form" and the use of "..in this search form" is not clear. Is search form a synonym of "search space"? Please clarify.
- (Modulared sub-tree discovery) "The complete from-discovery algorithm, ". There is a typo in "from-discovery". Please fix.
- (Results) How many expressions were evaluated in each algorithm? Is this the same number for all algorithms? This is very important to assess the computational complexity of the algorithm proposed. Please clarify.

---

> ### Author Response · Authors · 2023-11-17
> **Reply to Reviewer XG8a (Part 1)**
>
> We sincerely thank the reviewer for the constructive comments and suggestions, which are very helpful for improving our paper. The revisions have been incorporated in the revised manuscript marked in red.
>
> **Q1. Clarity and presentation of the paper.**
>
> **Reply:** Thank you for this comment. We have followed the reviewer’s suggestion and thoroughly revised the manuscript. Here are some examples of revision (more details are found in the revised manuscript):
>
> To better illustrate the novelty of our proposed model, an additional paragraph highlighting the motivation and novelty of the Modulated Sub-tree Discovery (MSDB) has been included.
>
> - *We introduced a new modulated sub-tree discovery (MSDB) block, the critical component of the proposed RSRM model, to heuristically learn and define new math operators to improve the representation ability of underlying math expression trees. Here, we would like to emphasize that MSDB addresses a crucial observation that models often struggle to generate complete expressions but excel in capturing certain components. For instance, NGGP may discover an expression like $x^4-x^3+\cos(y)+x-1$, while the ground truth is $x^4-x^3-0.5y^2+x$. Notably, it successfully recovers the simplified expression $x-0.5y^2$ with the same distribution. To this end, MSDB offers a new alternative to simplify expressions by subtracting specific components in the context of a sub-tree, as exemplified by the subtraction of $x^4-x^3$ in the aforementioned case.*
>
> Given the brevity of the description of the workflow in the previous version, we have revised the caption of Figure 1.
>
> - *RSRM comprises several steps. First, the data is input into the data pre-processing module to determine the parity. Each epoch, the process initiates with the utilization of the MCTS module to generate expressions (Step ①). Subsequently, these expressions undergo the addition of forms and reward calculation (Step ②). The expressions are then refined through GP (Step ③), leading to more accurate expressions (Step ④). Following this, the equation forms are integrated and rewards are calculated (Step ⑤). This iterative cycle, involving MCTS and GP, continues for several rounds, ultimately resulting in the renewal of forms based on the most proficient expressions (Step ⑥).*
>
> In the absence of a comprehensive description of the design of the three sub-tree expression forms, an additional paragraph has been added to explicate the genesis of these forms.
>
> - *In fact, the three forms (namely, ${\mathcal{A} + f(x)}$, $\mathcal{A}\times f(x)$, $\mathcal{A}^{f(x)}$) are general and applicable to express sub-tree structures of any generic formula. Our approach involves the establishment of these forms based on the initial token of the expression tree, because the root of an expression tree serves as a focal point, indicating the primary operation or function in the expression. Thus, we can separate the sub-tree forms based on it.  Specifically, if the first token corresponds to addition ($+$) or subtraction ($-$), the method proceeds to learn the generation of the left and right sides of the respective operators. Similarly, for tokens such as multiplication ($\times$), division ($\div$), or exponentiation (\^{}), a similar procedure is followed. In the case of unary expressions, such as trigonometric functions ($\sin$ and $\cos$), the MCTS and GP models effortlessly derive the complete expression. Therefore, while our method involves a degree of empirical design in identifying the sub-tree expression forms, it possesses a universal nature.*
>
> In addition, here are some examples of the revision of uncleared sentences:
>
> - *Symbolic Regression (SR) can be defined as the task of automatically distilling equations from limited data.*
>
> - *However, there still exist bottlenecks that current methods struggle to break when the expression we need to explore tends toward infinity and especially when the underlying math formula is intricate.*
>
> - *In the early years, polynomial analysis was commonly used to fit expressions.*
>
> - *By incorporating double Q-learning into MCTS, we effectively balance exploration and exploitation of SR tasks*
>
> - *The objective of SR can be transformed into the generation of an optimal expression tree.*
>
> We believe these revisions significantly enhance the clarity of our manuscript, making it easier to follow. We would like to take this opportunity to thank the reviewer for the careful reading and putting forward these detailed and constructive comments.

---

> ### Author Response · Authors · 2023-11-17
> **Reply to Reviewer XG8a (Part 2)**
>
> **Q2. The functionality of the interpolation method and its relation to data pre-processing.**
>
> **Reply:** Thanks for your comment. We employ an interpolation method to generate $f(-x)$ and directly compare it with $f(x)$ for each data point, establishing the presence of odd, even, or neither symmetry within the function. The determination of parity remains consistent throughout and precedes the search step, remaining unchanged thereafter. Hence, this meticulous procedure is a fundamental component of our pre-processing stage.
>
> **Q3. The meaning of “performing well” and details about the selection of expression forms.**
>
> **Reply:** Thank for your suggestion. Firstly, performing well means greater reward.
>
> Next, we give details of selecting expression forms: (1) In the form of multiplication ($\mathcal A \times f(x)$) or exponentiation ($\mathcal A^{f(x)}$), we select the best expression as the chosen form. (2) In the case of addition ($\mathcal A + f(x)$), we opt for expressions with a reward greater than $k_p \times \max\{\text{reward}\}$.Then those expressions are split and count each part. Additionally, we select components whose occurrences satisfy $o > k_s \times \max\{o\}$. The values of $k_s$ and $k_p$ are provided in Appendix A.1. For instance, if $x^5$ occurs 50 times, $-x^4$ occurs 4 times, and $x^3$ occurs 11 times, we finds the form as $x^5 - x^3 + f(x)$. These thresholds remain consistent across all datasets.
>
> **Q4. The unclear part of the form-discovery algorithm.**
>
> **Reply:** Thank for your notice. Please see our explanation as follows.
>
> > "This approach effectively reduces the difficulty associated with specific equations." This statement is not clear.
>
> For example, obtaining an expression like $x^4 - x^3 - 0.5y^2 + x$ directly can be challenging. However, by identifying its components such as $x^4$ and $-x^3$, we construct the form $x^4 - x^3 + x + f(x)$ to alleviate the difficulty of searching for the term $-0.5y^2$.
>
> > The concept of "search form" and the use of "..in this search form" is not clear. Is search form a synonym of "search space"? Please clarify.
>
> The term “search form” denotes the continuous exploration based on a predefined expression structure. For instance, in the form $e^x + f(x)$, we regard $e^x$ as an integral part of the overall expression. Therefore, if the outcome of the MCTS is $x + x^2$, the resultant expression would be $e^x + x + x^2$. Our calculations for RMSE and reward are based on this resultant expression.
>
> The form ($\mathcal A + f(x)$) considers that the entire expression is constructed by the concatenation of terms through sign addition ($+$) or subtraction ($-$). Consequently, if expressions such as $e^x - x$ and $e^x + x$ yield favorable results, it suggests that the overall expression may be of the form $e^x + f(x)$.
>
> Similarly, when the form($\mathcal A \times f(x)$) involves concatenation through multiplication ($\times$) or division ($\div$), the evaluation of expressions like $1.57e^x$ and $1.56e^x + x$ helps identify that the expression could be of the form $e^x \times f(x)$.
>
> **Q5. Number of expressions evaluated.**
>
> **Reply:** This is an excellent comment. In fact, there is a tradeoff between the recovery rate (accuracy) and the computational complexity of the model. We have performed a trade-off experiment in the manuscript illustrating the recovery rate in terms of the number of evaluations. Please see Appendix Section C.7 and Fig. S2. Here is the table of this experiment result for your reference.
>
> **Table.** Trade-off experiment: Average Number of Evaluations / Average Recovery Rate (%) of the Nyugen Benchmark over 100 parallel runs.
>
> | Name      | 5 epochs       | 15 epochs      | 25 epochs       | 35 epochs       | 45 epochs
> |-----------|----------------|----------------|-----------------|-----------------|------------------|
> | ALL       | ${15654}/{53}$ | ${27543}/{77}$ | ${41525}/{87}$  | ${55395}/{98}$  | ${58240}/{100}$
> | Nyugen-4  | ${28389}/{5}$  | ${41638}/{43}$ | ${54151}/{75}$  | ${58483}/{100}$ | ${76216}/{100}$
> | Nyugen-5  | ${25084}/{0}$  | ${57033}/{33}$ | ${77075}/{80}$  | ${118661}/{83}$ | ${149868}/{100}$
> | Nyugen-11 | ${20932}/{41}$ | ${36487}/{62}$ | ${67237}/{88}$  | ${77757}/{100}$ | ${84902}/{100}$
> | Nyugen-12 | ${26345}/{0}$  | ${65122}/{13}$ | ${123571}/{41}$ | ${175970}/{89}$ | ${207889}/{100}$
>
> Please do not hesitate to let us know if you have any further questions. Thank you very much.

---

> > ### Comment · Reviewer_XG8a · 2023-11-21
> > **Rebuttal Follow-up**
> >
> > I acknowledge that I have read the answers from the authors and the other reviewers. I appreciate the authors' efforts to improve the paper. I will maintain my positive evaluation.

---

> > > ### Author Response · Authors · 2023-11-21
> > > **Thanks for your positive feedback**
> > >
> > > We would like to take this opportunity to thank the reviewer for the positive feedback as well as these detailed and constructive comments.

---

### Official Review · Reviewer_AFH1 · 2023-10-30

**Soundness:** 3 good
**Presentation:** 2 fair
**Contribution:** 3 good
**Rating:** 8
**Confidence:** 3

**Summary:**

This paper proposes a new symbolic regression discovery method that uses Monte Carlo Tree Search for exploration, Double Q-Learning for exploitation, and a sub-tree discovery block to capture equation decompositions. All three components combined in their method named Reinforcement Symbolic Regression Machine, appear to achieve state-of-the-art results on a diverse range of standard symbolic regression benchmarks.

**Strengths:**

* The proposed approach appears novel and is well-placed in the relevant literature. The introduction does an excellent job of introducing the related work in a concise setting.
* The paper is well-written and easy to follow.
* The approach was empirically demonstrated to achieve state-of-the-art performance across the existing SR benchmark tasks and problem sets.
* The appendix is detailed and extensive, adding further empirical evidence to the core claims.

**Weaknesses:**

* There are no theoretical results to verify the proposed approach. However, the method does obtain strong empirical performance.
* Page 3, top. $c$ is undefined. It could perhaps be helpful to state what $c$ is in the UCT algorithm or provide a reference to either another paper or a forward reference to where it is defined.
* Page 5, "we find that Gaussian random numbers with a unit mean and variance provide more effective initial values for optimization.". There seems to be no empirical evidence in the paper supporting this. Although you may have results for this, it could be helpful for the reader if you included an additional Appendix experiment and linked it to this statement as the empirical evidence for this.
* There are no error bars for the main results tables, Table 1, Table 2, and Table 3.
* In Table S7, only 12 AI Feynman equations are used, whereas the original AI Feynman paper proposes 100 AI Feynman equations. Why were these 12 AI Feynman equations used, and is it possible to provide results for the complete set of AI Feynman equations?


Typos:
* Page 5: Splitting by Addition paragraph: "Expand" -> "Expanding"
* Page 5: Splitting by Addition paragraph: "Split" -> "Splitting"
* Page 5: Splitting by Addition paragraph: "converts" -> "convert"
* Page 5: Splitting by Addition paragraph: ", then transforms" -> ", and then transforms"

**Questions:**

* Can you define $c$ in the UCT algorithm in the text when you introduce it?
* Can you provide empirical evidence for the statement of "we find that Gaussian random numbers with a unit mean and variance provide more effective initial values for optimization.", perhaps in an additional appendix?
* Can you highlight why only 12 AI Feynman equations were used and possibly include results for all the AI Feynman equations in the AI Feynman problem set?

---

> ### Author Response · Authors · 2023-11-17
> **Reply to Reviewer AFH1 (Part 1)**
>
> We sincerely thank the reviewer for the constructive comments and suggestions, which are very helpful for improving our paper. The revisions have been incorporated in the revised manuscript marked in red.
>
> **Q1. Theoretical Analysis.**
>
> **Reply:** Thanks for this comment. Since primary part of the search method we employed in our proposed RL model is the UCT-based MCTS, the theoretical analysis (e.g., convergence, guarantees) of this algorithm can be found in literature such as Shah et al. (2019). We have clarified this in our revised manuscript.
>
> **Q2. Definition of $c$ in the UCT algorithm.**
>
> **Reply:** Thanks for the reviewer’s comment. $c$ typically represents the exploration-exploitation tradeoff parameter. The exploration-exploitation tradeoff is a fundamental challenge in decision-making under uncertainty. In the context of UCT, it refers to the balance between exploring new possibilities (potentially discovering a better move or solution) and exploiting known information (favoring moves or solutions that seem promising based on existing knowledge).
>
> Choosing an appropriate value for $c$ is crucial, as it influences the algorithm's behavior. A higher $c$ encourages more exploration, while a lower $c$ emphasizes exploitation. The optimal value for $c$ often depends on the specific problem or environment being addressed. It may need to be tuned through experimentation for different scenarios. In this paper , we take $c = \sqrt{2}$.
>
>
> **Q3. Experiment on initial value of constant optimization.**
>
> **Reply:**  This is an excellent remark. We conducted an experiment considering different initialization approaches for constant optimization. Specifically, we explored seven methods shown as follows (Case 1 – Case 7).
>
> Case 1: Initializing constants with a vector of ones (the initialization method in former models).
> Case 2: Initializing constants with a vector of random uniform values between 0 and 1.
> Case 3: Initializing constants with a vector of random uniform values between 0.5 and 1.5.
> Case 4: Initializing constants with a vector of random Gaussian values with a mean of 0 and standard deviation of 1.
> Case 5: Initializing constants with a vector of random Gaussian values with a mean of 1 and standard deviation of 1.
> Case 6: Initializing constants with a vector of random Gaussian values with a mean of 1 and standard deviation of 0.5.
> Case 7: Initializing constants with a vector of random Gaussian values with a mean of 0 and standard deviation of $\frac{1}{3}$.
>
> We evaluated the recovery rate of each expression using different constant initialization methods across diverse benchmarks, employing ranges of input values such as [0,1], [-1,1], [0,10], [-10,10], [0,50], [-50,50], and data sizes of 20 or 500. Notably, Table 1 demonstrates that the average recovery rates across all initializing methods are remarkably close.
>
> However, examining the distribution of expression recovery rates below 50%, 10%, and 0%, it becomes apparent that the method employing a vector of ones exhibits the highest percentage from Table 2. This observation indicates that the Case 1 initialization method (with the vector of all ones) has the highest number of expressions unable to converge once across 100 parallel runs.
>
> We have added a section in the Appendix to discuss this (see Appendix Section C.6).

---

> ### Author Response · Authors · 2023-11-17
> **Reply to Reviewer AFH1 (Part 2)**
>
> **Table 1.** Average Recovery Rate  of the Const Optimization Benchmark over 100 parallel runs
>
> | expression                                              | 1          | 2          | 3          | 4          | 5          | 6          | 7          |
> | ------------------------------------------------------- | ---------- | ---------- | ---------- | ---------- | ---------- | ---------- | ---------- |
> | $(30.0 x_1 x_2 x_3)/((x_1 - 10.0)x_2 ^ 2)$              | **34.2%**  | 13.3%      | 35.7%      | 14.8%      | 20.7%      | 24.3%      | 24.1%      |
> | $0.3x_1\sin(6.28x_1)$                                   | 33.6%      | 33.8%      | 33.6%      | 33.7%      | 33.7%      | 33.6%      | **34.3%**  |
> | $0.48x_1 ^ 4 + 3.39x_1 ^ 3 + 2.12x_1 ^ 2 + 1.78x_1$     | **98.0%**  | 92.3%      | 97.9%      | 92.9%      | 95.0%      | 97.5%      | 96.2%      |
> | $0.7x_1 ^ 3 - 1.7x_1$                                   | **100.0%** | 54.1%      | 92.2%      | 52.9%      | 76.1%      | 83.3%      | 87.9%      |
> | $1.35x_1 ^ 2 + 5.5\sin((x_1 - 1) ^ 2)$                  | **100.0%** | **100.0%** | **100.0%** | **100.0%** | **100.0%** | **100.0%** | **100.0%** |
> | $1.5\exp(x_1) + 5.0\cos(x_1)$                           | 87.7%      | 93.8%      | **94.6%**  | 88.1%      | 92.3%      | 94.5%      | 93.6%      |
> | $2.5  x_1 ^ 4 - 1.3  x_1 ^ 3 + 0.5  x_1 ^ 2 - 1.7  x_1$ | **98.3%**  | 97.5%      | 96.3%      | 97.0%      | 94.7%      | 97.7%      | 97.4%      |
> | $2.7  x_1 ^{x_2}$                                       | 99.2%      | 88.2%      | 98.5%      | 79.3%      | 89.3%      | 95.8%      | **100.0%** |
> | $3.39  x_1 ^ 3 + 2.12  x_1 ^ 2 + 1.78  x_1$             | 97.2%      | **97.5%**  | **97.5%**  | 96.8%      | 95.9%      | 96.9%      | 97.4%      |
> | $6.0  \sin(x_1)  \cos(x_1)$                             | **100.0%** | **100.0%** | **100.0%** | **100.0%** | **100.0%** | **100.0%** | **100.0%** |
> | $8.0  x_1 ^ 3 - 8.0  x_1 ^ 2 + 15.0  x_1$               | 81.0%      | 83.0%      | 81.4%      | 83.2%      | 83.1%      | 82.9%      | 82.5%      |
> | $8.0 / (2.0 + x_1 ^ 2 + x_2 ^ 2)$                       | **100.0%** | 99.8%      | **100.0%** | 82.0%      | 95.8%      | 99.6%      | 99.8%      |
> | $\log(x_1 + 1.4) + \log(x_1 ^ 2 + 1.3)$                 | **100.0%** | 53.4%      | 78.4%      | 50.6%      | 56.0%      | 70.4%      | 97.1%      |
> | $\sin(1.5  x_1)  \cos(0.5  x_2)$                        | 35.1%      | **47.7%**  | 38.7%      | 42.8%      | 42.7%      | 44.3%      | 42.3%      |
> | $\sin(1.5  x_1) + \sin(0.5  x_2 ^ 2)$                   | 35.3%      | **47.9%**  | 43.4%      | 42.6%      | 43.8%      | 45.6%      | 46.3%      |
> | $\sin(x_1 ^ 2)  \cos(x_1) - 0.75$                       | **100.0%** | **100.0%** | **100.0%** | **100.0%** | **100.0%** | **100.0%** | **100.0%** |
> | $\sqrt{1.23  x_1}$                                      | **100.0%** | **100.0%** | **100.0%** | 99.5%      | **100.0%** | **100.0%** | **100.0%** |
> | $x_1 ^ {0.6} + x_2 ^ {1.5} - x_2 - x_1$                 | 88.0%      | 83.2%      | **90.3%**  | 75.5%      | 83.8%      | 88.3%      | 88.7%      |
> | **average**                                             | 81.06%     | 76.19%     | 80.90%     | 73.46%     | 77.17%     | 79.74%     | **81.08%** |
>
>
>
> **Table 2.** Percentage of expressions under a certain recovery rate about the Const Optimization Benchmark over each initializing method
>
> | recovery rate | 1      | 2          | 3      | 4      | 5      | 6          | 7          |
> | ------------- | ------ | ---------- | ------ | ------ | ------ | ---------- | ---------- |
> | **<50%**      | 18.04% | 25.26%     | 18.04% | 26.29% | 19.07% | 17.53%     | **17.01%** |
> | **<10%**      | 15.98% | **11.34%** | 13.92% | 11.86% | 11.86% | **11.34%** | 12.37%     |
> | **0%**       | 7.22%  | 4.12%      | 6.19%  | 3.61%  | 4.64%  | 3.61%      | **3.09%**  |

---

> ### Author Response · Authors · 2023-11-17
> **Reply to Reviewer AFH1 (Part 3)**
>
> **Q4. The AIFeynman Benchmark results.**
>
> **Reply:** Thanks for your question. The selection of 12 AI Feynman equations was based on a stratified representation of difficulty levels for symbolic regression. The chosen expressions include easy ones, such as $x_1x_2$ and $\frac{3}{2}x_1x_2$; medium complex expressions like $x_1x_2x_3\sin(x_4)$ and $x_1x_2+x_3x_4+x_5x_6$; medium-hard expressions such as $\frac12x_1(x_2^2+x_3^2+x_4^2)$ and $x_1x_2x_3(\frac{1}{x_4}-\frac{1}{x_5})$; and hard expressions like $\frac{x_1x_2x_3}{(x_4-x_5)^2+(x_6-x_7)^2+(x_8-x_9)^2}$ and $1+\frac{x_1x_2}{1-x_1x_2/3}$. This selection for illustration provides a comprehensive coverage of the AIFeynman benchmark.
>
> Despite we only show the detailed results for the above 12 selection equations, the complete recovery results of the AIFeynman benchmark are summarized in Figure 3b. Notably, our model achieves a SOTA 80% average recovery rate, outperforming other existing representative baseline models and indicating its proficiency in capturing the underlying mathematical structures across a diverse range of expressions.
>
>
> **Q5. No error bars on the main results.**
>
> **Reply:** Thanks for your suggestion. In fact, since the recovery rates of individual equations within the same benchmark dataset do not follow the Gaussian distribution, the application of an error bar denoting the confidence range may not be suitable in this context. This is generally admitted in the symbolic regression community. However, we provide a table (see Table 3 below) outlining the 95% confidence range for each method within each benchmark for your reference.
>
> **Table 3.** 95% confidence range of recovery rate (%) for each method within each benchmark
>
> | Benchmark | OURS | SPL | uDSR | NGGP | DSR | GP |
> |:----:|:----:|:----:|:----:|:----:|:----:|:----:|
> | Nguyen | **100.00$\pm$0.0** | 93.50$\pm$11.7 | 87.58$\pm$13.6 | 91.75$\pm$13.0 | 83.58$\pm$18.5 | 44.42$\pm$22.1 |
> | Nguyen$^c$ | **100.00$\pm$0.0** | 69.57$\pm$35.2 | 60.71$\pm$33.2 | 99.14$\pm$1.2 | 70.43$\pm$35.7 | 8.43$\pm$15.6 |
> | Livermore | **97.95$\pm$4.0** | 51.00$\pm$16.9 | 81.05$\pm$14.6 | 80.68$\pm$14.0 | 50.41$\pm$15.7 | 50.77$\pm$20.4 |
> | R | **68.83$\pm$28.9** | 0.00$\pm$0.0 | 16.50$\pm$26.2 | 26.67$\pm$31.1 | 0.00$\pm$0.0 | 0.00$\pm$0.0 |
> | AIFeynman | **73.50$\pm$24.1** | 66.42$\pm$27.8 | 63.50$\pm$26.0 | 67.25$\pm$27.4 | 66.42$\pm$27.8 | 60.58$\pm$25.7 |
>
> Please do not hesitate to let us know if you have any further questions. Thank you very much.

---

> ### Author Response · Authors · 2023-11-21
> **Follow up on our response to Reviewer AFH1**
>
> Dear Reviewer AFH1,
>
> We extend our sincere gratitude for dedicating your valuable time to thoroughly review our paper and provide insightful comments. With only **two days** remaining in the Author-Review Discussion period, we want to ensure that we've satisfactorily addressed all your concerns. If there are any remaining questions or unresolved matters, please don't hesitate to reach out. We are glad to provide further clarification or make necessary revisions.
>
> Best regards,
>
> The Authors

---

> > ### Comment · Reviewer_AFH1 · 2023-11-22
> > **Reviewer AFH1 Response**
> >
> > Thank you for addressing all my listed constructive comments, suggestions, and concerns. I particularly find the new experiment on the initial value of constant optimization insightful. Does this new experiment suggest that Case 1, using an initialization method of constants with a vector of ones, is the best approach? I have increased my score to reflect my updated view. I would also encourage authors in the camera-ready to make their source code available for others to build upon their work.

---

> > > ### Author Response · Authors · 2023-11-22
> > > **Thanks for your positive feedback**
> > >
> > > We extend our gratitude to the reviewer for your positive feedback and the insightful, constructive comments provided.
> > >
> > > In our recent Constant Optimization experiment, the results revealed closely aligned average recovery rates across all initializing methods. However, case one prominently exhibited the highest percentage of recovery rates below 10% or equal to 0%, indicating a greater likelihood of failure despite numerous attempts.
> > >
> > > For instance, the expression $\sin(1.5x_1)+\sin(0.5x_2^2)$ couldn't be retrieved using case 1 by $\sin(C x_1)+\sin(C x_2^2)$, despite it is the accurate formulation. Conversely, case 7 efficiently found this expression within two or three attempts. Therefore, our favorite method is **case 3** or **case 7** due to their superior average performance.
> > >
> > > If you require further clarification or information, please don't hesitate to contact us. We greatly appreciate your valuable assistance.

---

> > > > ### Comment · Reviewer_AFH1 · 2023-11-22
> > > > **Reviewer AFH1 Response**
> > > >
> > > > Thank you again for this insightful response. Can you also release the source code upon publication of the paper to reproduce the results?

---

> > > > > ### Author Response · Authors · 2023-11-22
> > > > > **Thanks for your positive comment**
> > > > >
> > > > > We're pleased to receive your positive feedback. Certainly, upon the paper's publication, we will make the source code publicly available.

---

### Official Review · Reviewer_d71r · 2023-11-01

**Soundness:** 2 fair
**Presentation:** 3 good
**Contribution:** 2 fair
**Rating:** 6
**Confidence:** 4

**Summary:**

This paper studies the problem of discovering math equations in real-world complex systems. The authors propose a Reinforcement Symbolic Regression Machine (RSRM) that masters the capability of uncovering complex math equations from scarce data.  Experiments demonstrate the proposed method outperforms baselines on various benchmarks.

**Strengths:**

1.	The paper is well-written and easy to follow.
2.	Experiments demonstrate the proposed method outperforms baselines on various benchmarks.

**Weaknesses:**

1.	The novelty of the proposed method is unclear. The proposed method seems to be a simple combination of existing methods, including the monte carlo tree search (MCTS) algorithm, double q-learning method, and genetic programming method.
2.	The motivation and advantages of the proposed method over previous work are unclear.
3.	The authors claim that they use a MCTS agent for exploration. However, I found that they use greedy selection rather than exploration based on the upper confidence bound, which is a key component for efficient exploration in MCTS. Thus, whether the MCTS agent can explore the environment efficiently is unconvincing.
4.	The proposed modulated sub-tree discovery incorporate three specific search forms into the algorithm. However, the three forms may be too specific to be generally applicable to complex real-world problems.
5.	The authors evaluate the generalization ability of their method in a toy environment. It would be more convincing if the authors could conduct the generalization experiments on more complex real-world problems.

**Questions:**

Please refer to Weaknesses for my questions.

---

> ### Author Response · Authors · 2023-11-16
> **Reply to Reviewer d71r (Part 1)**
>
> We sincerely thank the reviewer for the constructive comments and suggestions, which are very helpful for improving our paper. The revisions have been incorporated in the revised manuscript marked in red.
>
> **Q1. Novelty, motivation and advantages of the model.**
>
> **Reply:** Thanks for this question. Firstly, we would like to mention that existing symbolic regression (SR) methods generally struggle with generating lengthy and complex equations, and are faced with issues related to overfitting, e.g., poor generalizability. We aimed to develop a new model (aka., RSRM) to overcome these critical challenges, which has shown SOTA performance as demonstrated over extensive examples on over 200 symbolic test formulas.
>
> Secondly, we do not agree with the reviewer’s comment on “*a simple combination of existing methods*”. The coordination of different methods or blocks requires a **rational** design (see the workflow in Figure 1) which largely affects the balance between exploration and exploitation. A simple combination would not work. In addition, we introduced a **new** modulated sub-tree discovery (MSDB) block, the critical component of the proposed RSRM model, to heuristically learn and define new math operators to improve the representation ability of underlying math expression trees. Here, we would like to emphasize that MSDB addresses a crucial observation that models often struggle to generate complete expressions but excel in capturing certain components. For instance, NGGP may discover an expression like $x^4-x^3+\cos(y)+x-1$, while the ground truth is $x^4-x^3-0.5y^2+x$. Notably, it successfully recovers the simplified expression $x-0.5y^2$ with the same distribution. To this end, MSDB offers a new alternative to simplify expressions by subtracting specific components in the context of a sub-tree, as exemplified by the subtraction of $x^4-x^3$ in the aforementioned case.
>
> Therefore, the **novelty** of the proposed model is three-fold: *(1)* Using double Q-learning in junction with MCTS effectively reduces redundant search thus improving the exploration efficiency and efficacy. *(2)* The proposed MSDB block for discovering new sub-tree expression forms takes the divide-and-conquer concept and could significantly improve the overall search performance of the RSRM model. In particular, MSDB can handle equations with symmetry (reducing the complexity), and assist in dealing with long equations by identifying common patterns and defining new math operators on the fly. *(3)* The proposed RSRM model demonstrates clear superiority over many representative baseline models and yields the SOTA SR results, which offers a new approach to tackle very complex SR challenges.
>
>
> **Q2. Exploration efficiency of MCTS.**
>
> **Reply:** This is a good question. In fact, the UCT is employed during the MCTS simulation while the greedy selection of the maximum reward is only applied to choose the optimal expression tree (as shown in Algorithm 1). We have tested, when we designed the model, that greedy selection of the maximum reward performs better that the case of the highest UCT score in the context of SR tasks.
>
> Note that the maximum UCT score-based greedy selection method often selects nodes with a lower possibility to generate non-meaningful equations. For instance, it may prefer selecting $\sin$ nodes across all instances range over $\log$ nodes, because the output of $\sin$ is  within the [-1,1] range, and the latter might yield NaN for negative values. However, this preference doesn't align with our objective, as the $\log$ token could potentially be the optimal choice. Hence, we opt to only consider the best expression (with the maximum reward) to determine which node to use in the selection step.

---

> ### Author Response · Authors · 2023-11-16
> **Reply to Reviewer d71r (Part 2)**
>
> **Q3. Expression forms that may restrict the applicability of the method to real-world problems.**
>
> **Reply:** Thanks for the reviewer’s excellent comment. In fact, the three forms (namely, ${\mathcal{A} + f(x)}$, $\mathcal{A}\times f(x)$, $\mathcal{A}^{f(x)}$) are general and applicable to express sub-tree structures of any generic formula. Our approach involves the establishment of these forms based on the initial token of the expression tree, because the root of an expression tree serves as a focal point, indicating the primary operation or function in the expression. Thus, we can separate the sub-tree forms based on it.  Specifically, if the first token corresponds to addition ($+$) or subtraction ($-$), the method proceeds to learn the generation of the left and right sides of the respective operators. Similarly, for tokens such as multiplication ($\times$), division ($\div$), or exponentiation (\^{}), a similar procedure is followed. In the case of unary expressions, such as trigonometric functions ($\sin$ and $\cos$), the MCTS and GP models effortlessly derive the complete expression. Therefore, while our method involves a degree of empirical design in identifying the sub-tree expression forms, it possesses a universal nature.
>
>
> **Q4. The generalization experiment on more complex problems.**
>
> **Reply:** Thanks for your comment and suggestion. The reason why we tested our model by discovering a surrogate formula to approximate the cumulative density function (CDF) of a normal distribution is because this equation lacks an explicit elementary expression. Hence, it's impossible to obtain an exact solution through SR. Learning this function requires balancing between training error and generalization error, demanding a stronger ability to generalize. Moreover, only half of the curve data is provided, necessitating strong generalization skills from the model to extrapolate and accurately fit the missing portion of the curve. This calls for the proficiency of the model in capturing the distribution of the entire curve. In summary, this is in fact not a simple SR case, in which many existing SR methods fail.
>
> Indeed, we acknowledge the importance of assessing the generalization ability of our model in more complex real-world scenarios. To address this, we conducted tests on the SRBench datasets, a benchmark comprising complex real-world problems with added noise. The results (see Figure 3b and Appendix Figure S3) demonstrate the superior performance of our model compared to other models, including UDSR, which has been recognized as the leading model in the preceding year.

---

> > ### Comment · Reviewer_d71r · 2023-11-22
> > **Thanks for the authors’ rebuttal**
> >
> > Thanks for the authors' response to address most of my major concerns. I have raised my score to 6.

---

> > > ### Author Response · Authors · 2023-11-22
> > > **Thanks for your positive feedback**
> > >
> > > We would like to take this opportunity to thank the reviewer for the positive feedback as well as these detailed and constructive comments.

---

> ### Author Response · Authors · 2023-11-21
> **Follow up on our response to Reviewer d71r**
>
> Dear Reviewer d71r,
>
> We extend our sincere gratitude for dedicating your valuable time to thoroughly review our paper and provide insightful comments. With only **two days** remaining in the Author-Review Discussion period, we want to ensure that we've satisfactorily addressed all your concerns. If there are any remaining questions or unresolved matters, please don't hesitate to reach out. We are glad to provide further clarification or make necessary revisions.
>
> Best regards,
>
> The Authors

---

### Official Review · Reviewer_jkfW · 2023-11-01

**Soundness:** 3 good
**Presentation:** 2 fair
**Contribution:** 3 good
**Rating:** 6
**Confidence:** 2

**Summary:**

This paper proposes Reinforcement Symbolic Regression Machine (RSRM), a symbolic regression method that combines reinforcement learning (RL), genetic programming (GP), and a novel modulated sub-tree discovery block. RSRM alternates between a reinforcement learning stage that searches for the optimal expression tree using double Q-learning and MCTS and a GP stage that refines the expression trees. At the end of each epoch, MSDS discovers new expression forms that reduce the search space for subsequent steps. RSRM demonstrates superior performance over baselines across a suite of symbolic regression datasets.

**Strengths:**

- The idea of combining RL and GP is a novel contribution. Using double Q-learning in junction with MCTS effectively reduces the search space. This is further enhanced by discovering new expression forms in MSDB.
- The method achieves strong performance across a suite of datasets, outperforming baselines in terms of expression recovery rate.
- The authors conducted extensive ablation studies to validate the effectiveness of each component in contributing to the overall performance.

**Weaknesses:**

- There is a limited set of hand-designed expression forms, which restricts the applicability of the method to other domains.
- It is a bit illusive how SR can be applied to practical domains.

**Questions:**

- Can you elaborate on each step in figure 1? The way it's presented now makes it hard to understand.
- What are some practical domains that SR is useful in? In particular, what are some scenarios where SR outperforms neural network regression?

---

> ### Author Response · Authors · 2023-11-16
> **Reply to Reviewer jkfW**
>
> We sincerely thank the reviewer for the constructive comments and suggestions, which are very helpful for improving our paper. The revisions have been incorporated in the revised manuscript marked in red.
>
> **Q1. Expression forms that may restrict the applicability of the method to other domains.**
>
> **Reply:** Thanks for the reviewer’s excellent comment. In fact, the three forms (namely, ${\mathcal{A} + f(x)}$, $\mathcal{A}\times f(x)$, $\mathcal{A}^{f(x)}$) are general and applicable to express sub-tree structures of any generic formula. Our approach involves the establishment of these forms based on the initial token of the expression tree, because the root of an expression tree serves as a focal point, indicating the primary operation or function in the expression. Thus, we can separate the sub-tree forms based on it.  Specifically, if the first token corresponds to addition ($+$) or subtraction ($-$), the method proceeds to learn the generation of the left and right sides of the respective operators. Similarly, for tokens such as multiplication ($\times$), division ($\div$), or exponentiation (\^{}), a similar procedure is followed. In the case of unary expressions, such as trigonometric functions ($\sin$ and $\cos$), the MCTS and GP models effortlessly derive the complete expression. Therefore, while our method involves a degree of empirical design in identifying the sub-tree expression forms, it possesses a universal nature.
>
>
> **Q2. Application of SR in practical domains and its superiority over neural network regression.**
>
> **Reply:** Thanks for this question. Symbolic regression (SR) has found broad applications across diverse fields, including discovery of fundamental physical laws [1, 2] or governing equations [3-5], modeling material constitutive relations [6], and TCP congestion control [7], among many others. For example, in the case of TCP congestion control, SR can be used to learn the control policy with better interpretability and generalization ability, which also exhibit advantage of computational speed in comparison with neural network regression models. In addition, when the training data is very limited, the neural network model tends to have the over-fitting issue, whereas SR can still produce a model with good generalizability.
>
>
> **Q3. Elaboration on each step in figure 1.**
>
> **Reply:** Thanks for your suggestion. We have added more detailed description of each step about Figure 1 in its caption, e.g.,
>
> “*First, the data is input into the data pre-processing module to determine the parity. Each epoch, the process initiates with the utilization of the MCTS module to generate expressions (Step ①). Subsequently, these expressions undergo the addition of forms and reward calculation (Step ②). The expressions are then refined through GP (Step ③), leading to more accurate expressions (Step ④). Following this, the equation forms are integrated and rewards are calculated (Step ⑤). This iterative cycle, involving MCTS and GP, continues for several rounds, ultimately resulting in the renewal of forms based on the most proficient expressions (Step ⑥).*”
>
> Please do not hesitate to let us know if you have any further questions. Thank you very much.
>
> **References:**
>
> [1] Silviu-Marian Udrescu and Max Tegmark. AI Feynman: A physics-inspired method for symbolic regression. Science Advances, 6(16): eaay2631, 2020.
>
> [2] Ziming Liu and Max Tegmark. Machine learning conservation laws from trajectories. Physical Review Letters, 126(18):180604, 2021.
>
> [3] Michael Schmidt and Hod Lipson. Distilling free-form natural laws from experimental data. Science, 324(5923):81–85, 2009.
>
> [4] Zhao Chen, Yang Liu, and Hao Sun. Physics-informed learning of governing equations from scarce data. Nature Communications, 12(1):6136, 2021.
>
> [5] Fangzheng Sun, Yang Liu, Jian-Xun Wang, and Hao Sun. Symbolic physics learner: Discovering governing equations via Monte Carlo tree search. International Conference on Learning Representations, 2023.
>
> [6] Yiqun Wang, Nicholas Wagner, and James M Rondinelli. Symbolic regression in materials science. MRS Communications, 9(3):793–805, 2019.
>
> [7] SP Sharan, Wenqing Zheng, Kuo-Feng Hsu, Jiarong Xing, Ang Chen, and Zhangyang Wang. Symbolic distillation for learned TCP congestion control. Advances in Neural Information Processing Systems, 35:10684–10695, 2022.

---

> > ### Comment · Reviewer_jkfW · 2023-11-20
> >
> > Thanks for the clarifications. I find the revised Figure 1 much easier to understand. It seems that the method is quite general and applies to a plethora of practical applications. Therefore, I will maintain my positive evaluation.

---

> > > ### Author Response · Authors · 2023-11-21
> > > **Thanks for your positive feedback**
> > >
> > > We would like to take this opportunity to thank the reviewer for the positive feedback as well as these detailed and constructive comments.

---

### Author Response · Authors · 2023-11-17
**Revised paper uploaded and response to reviewers’ comments posted**

Dear Reviewers:

We would like to thank you for your constructive comments, which are very helpful for improving our paper. We have posted the point-to-point reply to each question/comment raised by you and uploaded the revised version of our paper (with track changes marked in red). Please do feel free to let us know if you have any further questions.

Thank you very much.

Best regards,

The Authors of the Paper

---

### Meta-Review · Area_Chair_zXic · 2023-12-05

**Metareview:**

This paper proposes new improvements to the application of Monte-Carlo Tree Search (MCTS) to symbolic regression (SR) problems, especially over [1]. The general idea here is that a symbolic regression tree can be seen through the lens of MCTS, as the tree can grow / shrink based on search.

The new improvements consist of:
* Double Q-learning to improve exploration/exploitation tradeoffs
* Dealing with very long expressions (via observing symmetry and atomicizing subexpressions / subtrees)

Throughout the reviewing process, the paper obtained high (8, 6, 6, 6) scores, especially due to its high experimental performance. However, a common complaint was that the presentation could be significantly improved, which I agree. I strongly suggest that the authors review Sections 1-3, and highlight exactly what their contributions are, and rewrite Section 3.3 more cleanly.

[1] https://arxiv.org/abs/2205.13134

**Justification For Why Not Higher Score:**

The presentation issues make it very difficult to figure out whether any of the proposed improvements are significant (or how those improvements even work).

**Justification For Why Not Lower Score:**

Reviewers unanimously gave high scores especially due to the strong experimental results, and thus the paper at least warrants a poster accept.

---

### Decision · Program_Chairs · 2024-01-16

Accept (poster)